# Beyond One-Hot Labels: Semantic Mixing for Model Calibration

Haoyang Luo [1]  Linwei Tao [2]  Minjing Dong [1]  Chang Xu [2]

## Abstract

Model calibration seeks to ensure that models produce confidence scores that accurately reflect the true likelihood of their predictions being correct. However, existing calibration approaches are fundamentally tied to datasets of one-hot labels implicitly assuming full certainty in all the annotations. Such datasets are effective for classification but provides insufficient knowledge of uncertainty for model calibration, necessitating the curation of datasets with numerically rich ground-truth confidence values. However, due to the scarcity of uncertain visual examples, such samples are not easily available as real datasets. In this paper, we introduce calibration-aware data augmentation to create synthetic datasets of diverse samples and their ground-truth uncertainty. Specifically, we present **Calibration-aware Semantic Mixing (CSM)**, a novel framework that generates training samples with mixed class characteristics and annotates them with distinct confidence scores via diffusion models. Based on this framework, we propose calibrated reannotation to tackle the misalignment between the annotated confidence score and the mixing ratio during the diffusion reverse process. Besides, we explore the loss functions that better fit the new data representation paradigm. Experimental results demonstrate that CSM achieves superior calibration compared to the state-of-the-art calibration approaches. Our code is available here.

## 1. Introduction

Modern deep neural networks (DNNs) have achieved significant effectiveness in various vision tasks including image recognition, retrieval, and object segmentation (Zagoruyko, 2016; Gordo et al., 2016; Kirillov et al., 2023). Although

[1]Department of Computer Science, City University of Hong Kong [2]School of Computer Science, University of Sydney. Correspondence to: Minjing Dong <minjdong@cityu.edu.hk>.

*Proceedings of the 42nd International Conference on Machine Learning*, Vancouver, Canada. PMLR 267, 2025. Copyright 2025 by the author(s).

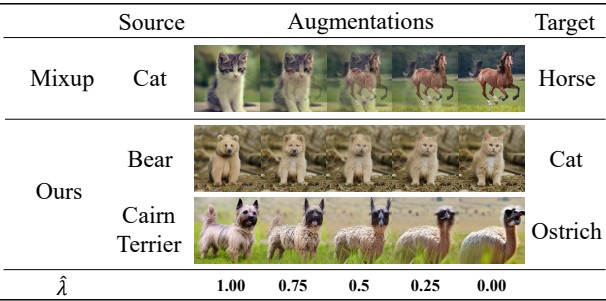

| | Source | Augmentations | Target |
|---|---|---|---|
| Mixup | Cat | | Horse |
| Ours | Bear | | Cat |
| | Cairn Terrier | | Ostrich |
| $\hat{\lambda}$ | | 1.00  0.75  0.5  0.25  0.00 | |

*Figure 1.* **Differences between Mixup and the Proposed Augmentation.** Samples with intermediate mixing coefficients exhibit simple overlapping pattern for Mixup. The proposed augmentation mixes semantics while preserving the object completeness, displaying enhanced image fidelity.

DNNs can achieve exceptional accuracy, they could be unreliable in real-world scenarios since the predictions are always over-confident (Mukhoti et al., 2020; Wei et al., 2022). This issue could mislead security-sensitive applications, such as autonomous systems, surveillance applications, medical diagnostics, *etc*. Model calibration (Vaicenavicius et al., 2019; Tao et al., 2023b), the process which refines the models' predicted confidence to be consistent with the actual probability of correct prediction, stands as the key to reliable vision models.

Various calibration techniques have been introduced, including post-hoc calibration (Platt et al., 1999; Guo et al., 2017; Tao et al., 2025), loss function designs (Mukhoti et al., 2020; Tao et al., 2023a), and regularization methods (Kumar et al., 2018; Krishnan & Tickoo, 2020). However, current methods rely on one-hot labeled data, which inaccurately assume uniform uncertainty across samples for confidence estimation. For instance, while a clear cat image and an ambiguous cat-dog hybrid image demand distinct confidence distributions, one-hot labels enforce identical certainty. This creates a critical gap that current datasets lack ground-truth uncertainty annotations necessary to teach models nuanced confidence distinctions. However, real-world collection of such data is infeasible due to scalability and ambiguity challenges. This necessitates us to create synthetically generated datasets with diverse ground-truth confidence annotations.

To generate samples with ground-truth uncertainty, a naive solution could be Mixup-based approaches (Yun et al., 2019; Kim et al., 2020; Uddin et al., 2021; Chen et al., 2022; Noh et al., 2023). By augmenting training data with pixel-interpolated samples and soft labels, DNNs are supposed to learn diverse confidence. However, such methods often produce low-fidelity samples due to unrealistic pixel fusion or fragmentary collage, which deviate significantly from the distribution of real-world data as shown in Figure 1. Consequently, these approaches struggle to generalize the learned calibration knowledge to real-world scenarios and hardly contribute to the calibration improvement (Wen et al., 2021; Maroñas et al., 2021; Wang et al., 2023a). Data augmentations have also been explored at test time (Hekler et al., 2023) to improve uncertainty estimation. While these post-hoc methods can mitigate over-confidence globally, they often fail to achieve precise calibration because they ignore miscalibration in the training with one-hot samples.

For successful calibration-aware data augmentation, realistic samples with different soft class posteriors are expected to be constructed, which remains unexplored in the field of calibration. In this paper, we introduce Calibration-aware Semantic Mixing (CSM), a data augmentation approach designed for model calibration that generates high-fidelity samples with semantic mixing. Specifically, we adopt pre-trained diffusion models to sample different sets of augmented images. Within the same set, the generated images are conditioned on the same latent noise but different soft class posteriors. Therefore, objects can be continuously transformed in the pixel layout from the instance of one class to another, which preserves the basic spatial completeness of objects to ensure image fidelity.

Besides these image samples, their corresponding soft labels should be annotated for learning. A naive solution is the mixing ratio during the diffusion reverse process. However, it can rarely indicate the class posterior precisely. To tackle this issue, we propose to reannotate the data by leveraging CLIP's visual features which have been jointly trained with massive language supervision (Radford et al., 2021), and are thus able to provide more precise class posteriors. By representing the optimized sample embeddings as interpolations of class prototypes, we can identify and eliminate class-specific biases in the estimated mixing coefficients, resulting in more accurate and effective annotations. Moreover, we highlight the problem of imbalanced fitting of augmented data when utilizing traditional loss functions for calibration through some theoretical analysis. We reveal that the training of augmented data with $\mathcal{L}_2$ loss intrinsically results in balanced learning and better calibration performance. Extensive experiments on various benchmarks and tasks demonstrate our method's strong effectiveness in accurately estimating confidence levels, achieving superior results for data-aware model calibration.

## 2. Related Work

### 2.1. Model Calibration

Network calibration (Guo et al., 2017) requires techniques to align model confidence with actual accuracy. Post-hoc methods adjust test-time parameters using a validation set to improve calibration. Notable techniques include Temperature Scaling (TS) (Guo et al., 2017), Histogram Binning (Zadrozny & Elkan, 2001), Beta calibration (Kull et al., 2017) and its extension to Dirichlet calibration (Kull et al., 2019). While simple, post-hoc methods are sensitive to distributional shifts (Ovadia et al., 2019). Training-time regularization methods involve explicit learning constraints and implicit loss functions. Label Smoothing (LS) (Müller et al., 2019) reduces output entropy by replacing one-hot targets with smoothened labels. Margin-based Label Smoothing (MbLS) (Liu et al., 2022) balances model discriminability and calibration by imposing a margin on logits. Implicit loss functions, such as those optimizing Expected Calibration Error (Karandikar et al., 2021; Kumar et al., 2018), Focal loss (Ross & Dollár, 2017; Mukhoti et al., 2020), Inverse Focal loss (Wang et al., 2021), and Mean Square Error (Hui & Belkin, 2021; Liang et al., 2024), are effective objectives to improve calibration. Data-aware calibration primarily involves perturbation methods (Tao et al., 2024) and Mixup-based methods (Zhang et al., 2018; Yun et al., 2019; Hendrycks et al., 2019), originally designed for improved generalization . These techniques have been found effective for calibration (Zhang et al., 2018; Pinto et al., 2022). RegMixup (Pinto et al., 2022) uses Mixup as a regularizer for cross-entropy loss, enhancing both accuracy and uncertainty estimation. RankMixup (Noh et al., 2023) incorporates ordinal ranking relationships among samples to reduce Mixup label bias. However, recent work (Wang et al., 2023a) highlights the limitation of simple convex combinations in Mixup methods.

### 2.2. Generative Data Augmentation

Generative methods, particularly Diffusion Models (DM), have recently gained popularity for data augmentation due to their high fidelity (He et al., 2022). Label-preserving augmentations based on DMs have been proposed to enhance image diversity. DA-Fusion (Trabucco et al., 2023) uses the SDEdit (Meng et al., 2021) technique to produce controlled variations of images within the same class. Diff-Mix (Wang et al., 2024) performs image translations by transferring foreground objects to a different class while preserving the background. In contrast, DiffuseMix (Islam et al., 2024) modifies the background with crafted prompts to enhance style diversity. To mix class concepts, MagicMix (Liew et al., 2022) combines SDEdit with specific prompts to generate semantically mixed images. De-DA (Chen et al., 2024) decouples and re-blends class's dependent and independent

parts to create one-hot or mixed-label augmentations.

# 3. Method

We provide preliminaries of model calibration and diffusion models in Section 3.1 and then introduce our proposed framework illustrated in Figure 2, regarding details of semantic mixing, reannotation, and learning objective for model calibration in Section 3.2 and Section 3.3.

## 3.1. Preliminaries

**Model Calibration** Considering an image classification task with a dataset $\{(\boldsymbol{x}_i, y_i)\}_{i=1}^N$, where $\boldsymbol{x}_i$ is a sample and $y_i$ is its true label, a DNN classifier $f_\theta$ predicts class probabilities $\boldsymbol{p}_i = \{p_i^k\}_{k=1}^K = \mathrm{softmax}(f_\theta(\boldsymbol{x}_i)) \in \Delta^K$, with $p_i^k$ representing the predicted probability for class $k$. Here, $\Delta^K = \{\boldsymbol{p} \in [0,1]^K, \sum_{k=1}^K p_k = 1\}$ represents the $K$-dimensional probability space. The predicted label for sample $i$ is

$$\hat{y}_i = \arg\max_k p_i^k. \quad (1)$$

The confidence score of a predicted label $\hat{y}_i$ is defined as $\hat{p}_i = p_i^{\hat{y}_i}$. Formally, a model is well-calibrated if the predicted confidence matches the true accuracy, *i.e.*, $\mathbb{P}(\hat{y}_i = y_i \mid p_i^{\hat{y}_i} = p)$ for $p \in [0,1]$. To measure network calibration, the Expected Calibration Error (ECE) is a commonly-used metric defined as $\mathbb{E}_{\hat{p}_i}[\mathbb{P}(\hat{y}_i = y_i \mid \hat{p}_i) - \hat{p}_i]$. Practically, due to the finite samples available, ECE is approximated as the weighted average of the absolute difference between accuracy and confidence across $M$ bins of grouped confidence values. Formally, ECE is defined as:

$$\mathrm{ECE} = \frac{1}{M} \sum_{m=1}^M |B_m| \cdot |\mathrm{Acc}(B_m) - \mathrm{Conf}(B_m)|, \quad (2)$$

where $\mathrm{Acc}(B_m)$ and $\mathrm{Conf}(B_m)$ denote the accuracy and average confidence in the $m$-th bin, and $|B_m|$ is the number of samples in $B_m$.

**Diffusion Models for Image Generation** Diffusion models are trained to generate realistic images via a gradual denoising process upon Gaussian noises. Their forward process gradually adds noise through a Markov chain with Gaussian transitions $q_{\mathrm{fwd}}(\boldsymbol{x}_i^{(t)}|\boldsymbol{x}_i^{(t-1)}) = \mathcal{N}(\boldsymbol{x}_i^{(t)}; \sqrt{\eta_t}\boldsymbol{x}_i^{(t)}, (1-\eta_t)\boldsymbol{I})$, where $\boldsymbol{x}_i^{(t)}$ is the noised variable of $\boldsymbol{x}_i^{(0)} = \boldsymbol{x}_i$ at step $t$ and $\eta_t$ is the noise schedule. The objective for training conditional denoising parameters $\phi$ is represented as $\min_\phi \mathbb{E}_{\boldsymbol{\epsilon}, \boldsymbol{x}, y, t} \|\boldsymbol{\epsilon} - \boldsymbol{\epsilon}_\phi^t(\boldsymbol{x}^{(t)}, y)\|$, where $\boldsymbol{\epsilon}_\phi$ is the predicted noise. For conditional image generation, the classifier-free guidance (CFG) (Ho & Salimans, 2022) is adopted to denoise images altering noise prediction as

$$\widetilde{\boldsymbol{\epsilon}}_\phi(\boldsymbol{x}^{(t)}, y) = (1+\omega)\boldsymbol{\epsilon}_\phi(\boldsymbol{x}^{(t)}, y) - \omega\boldsymbol{\epsilon}_\phi(\boldsymbol{x}^{(t)}, \varnothing), \quad (3)$$

where $\omega$ denotes the guidance strength.

## 3.2. Calibration-aware Data Augmentation

Existing calibration approaches (Guo et al., 2017; Mukhoti et al., 2020; Kumar et al., 2018; Müller et al., 2019) primarily focus on post-hoc and training techniques, while the training data for calibration is rarely considered. In fact, the training is always performed on the datasets with an ordinary structure, *i.e.*, the data pair of an image and its corresponding single label in the image classification task. Such annotated data exhibit zero uncertainty which can be ineffective for confidence calibration. To obtain diverse samples of ground-truth uncertainty values, we propose to tackle model calibration via a calibration-aware data augmentation scheme. Specifically, soft-labeled data are defined as $\{(\widetilde{\boldsymbol{x}}_a, \widetilde{\boldsymbol{q}}_a)\}_{a=1}^A$, where $A$ is the total number of augmented samples and $\{\widetilde{q}_a^k\}_{k=1}^K \in \Delta^K$ can be the non-binary confidence value for each class $k$ of sample $\widetilde{\boldsymbol{x}}_a$. We expect such data to provide confidence knowledge via soft label $\widetilde{\boldsymbol{q}}_a$ instead of single class $y$ in order to calibrate the model.

A straightforward calibration scheme generating soft-labeled data is the mixup (Zhang et al., 2018) augmentation, which generates fused samples and labels from random training pairs as $\widetilde{\boldsymbol{x}} = \lambda\boldsymbol{x}_i + (1-\lambda)\boldsymbol{x}_j$ and $\widetilde{\boldsymbol{q}} = \lambda\boldsymbol{q}_i + (1-\lambda)\boldsymbol{q}_j$, where $\lambda \sim \mathrm{Beta}(\alpha, \alpha) \in [0,1]$ is a mixup coefficient, $\alpha > 0$ is a hyperparameter controlling the interpolation strength (Noh et al., 2023), and $\boldsymbol{q}_i$ is the one-hot vector of $y_i$. However, the direct utilization of mixup cannot contribute to the calibration improvement, as empirically found by several studies (Wen et al., 2021; Maroñas et al., 2021; Wang et al., 2023a). We mainly attribute this failure to the fact that mixup performs calibration learning on low-fidelity images due to simple convex combinations, which makes the learned calibration difficult to generalize to real-world data. Therefore, to establish an effective calibration-aware data augmentation, we propose to generate high-fidelity images with semantic mixing, which reflects the shifting process of class posteriors from one category to another.

We start by expressing the general problem as how to establish a vicinal distribution that can be sampled from, *i.e.*,

$$\widetilde{\boldsymbol{x}} \sim p_v(\widetilde{\boldsymbol{x}} \mid \boldsymbol{x}_i, \boldsymbol{x}_j),$$
$$\text{s.t. } (\boldsymbol{x}_i, \boldsymbol{x}_j) \sim p_{dual}(\boldsymbol{x}_i, \boldsymbol{x}_j \mid y_i, y_j), \quad (4)$$

where $p_{dual}$ is the joint distribution for $\boldsymbol{x}_i$ and $\boldsymbol{x}_j$ and $p_v$ is the vicinal distribution to be designed for semantic mixing.

To blend class semantics with soft labels, we propose a dual-conditioned sampling strategy via a conditional diffusion model explicitly optimized to estimate $p(\boldsymbol{x} \mid y)$, *i.e.*,

$$p_\phi(\boldsymbol{x} \mid \boldsymbol{q}^y) = \int_{\boldsymbol{x}^{(T)}} p_\phi(\boldsymbol{x} \mid \boldsymbol{q}^y, \boldsymbol{x}^{(T)}) p(\boldsymbol{x}^{(T)}) \mathrm{d}\boldsymbol{x}^{(T)}, \quad (5)$$

Specifically, we generate augmented samples $\{\widetilde{\boldsymbol{x}}^{\hat{\lambda}}\}$ via dif-

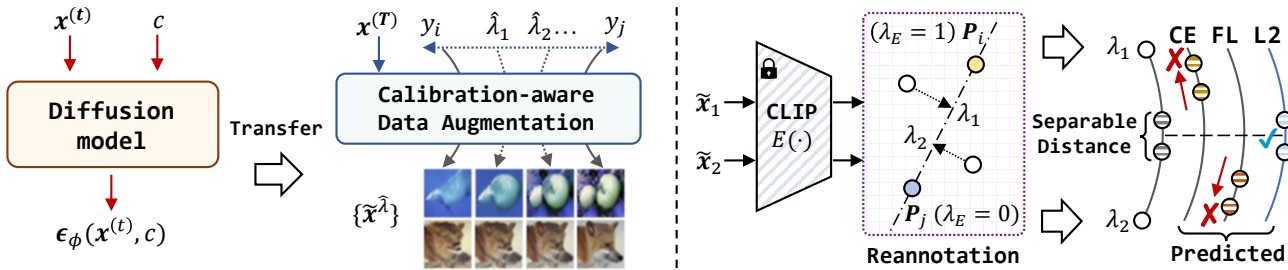

*Figure 2.* **The framework of the proposed Calibration-aware Semantic Mixing (CSM).** Left: Calibration-aware Data Augmentation exploits a pretrained diffusion model to ensure fidelity and coherence in generating conceptually mixed augmentations. Right: To further improve the annotation and fitting of class posteriors, a normalized reannotation scheme and the theoretically unbiased L2 loss are adopted to achieve balanced confidence calibration.

fusion reverse process as

$$\widetilde{x}^0 \sim p_\phi(x \mid q^{y_j}, x^{(T)}), \quad \widetilde{x}^1 \sim p_\phi(x \mid q^{y_i}, x^{(T)}),$$
$$\widetilde{x}^{\hat\lambda} \sim p_\phi(x \mid \widetilde{q}^{\hat\lambda}, x^{(T)}), \quad x^{(T)} \sim \mathcal{N}(x^{(T)}, 0, I), \quad (6)$$

where $\hat\lambda \in [0, 1]$. The augmentation is simultaneously conditioned on the mixed label $\widetilde{q}^{\hat\lambda} = \hat\lambda q^{y_i} + (1 - \hat\lambda) q^{y_j}$ and the sampled Gaussian noise $x^{(T)}$ at diffusion timestep $T$. The merits here are 3-fold: 1) the data fusion introduced at diffusion conditioning rather than pixel levels can significantly improve the image fidelity, which aligns with the real data distributions. 2) $\widetilde{x}^0$, $\widetilde{x}^1$, and $\widetilde{x}^{\hat\lambda}$ are correlated by conditioning on a common hidden state $x^{(T)}$, which controls the general image layout, maximizes the proximity of the augmentation series, and ensures unified visual composition for calibration learning. 3) conditional diffusion models are implicit class posterior estimators (Li et al., 2023) that can generalize to unseen conditions (Li et al., 2024), *i.e.*, mixed labels in our case. Therefore, $\widetilde{q}^{\hat\lambda}$ can be a naive solution to the label of augmented sample $\widetilde{x}^{\hat\lambda}$.

### 3.3. Calibrated Reannotation and Balanced Learning

While the proposed data augmentation scheme in Section 3.2 demonstrates various advantages for calibration learning, several potential issues arise when considering the characteristics of semantic mixing data pair $(\tilde{x}, \tilde{q})$. One issue lies in the annotation. Given a semantic-mixed image sample $\widetilde{x}^{\hat\lambda}$, it is necessary to provide an accurate annotation of its true class posterior. However, the annotated soft label $\widetilde{q}^{\hat\lambda}$ from diffusion models can hardly be equivalent to the true class posterior since diffusion models are also pretrained on the datasets with hard labels. In fact, we empirically found that the generated sample set $\{\widetilde{x}^{\hat\lambda}\}$ where $\hat\lambda \in [0, 1]$ shows high proximity, which strengthens the layout and object coherence due to the joint conditioning on $x^{(T)}$. As shown in Figure 4, $\hat\lambda$ may not accurately reflect the true class posterior due to non-linear semantic transitions in generated images. Another issue arises in the training phase,

as the existing loss functions are specifically designed for hard labels. We provide some theoretical analysis of the potential bias induced by popular loss functions used in the calibration field during the training of soft-labeled data. To tackle these issues, we introduce calibrated reannotation and balanced learning to ensure effective calibration learning.

**Calibrated Reannotation**   As analyzed, $\hat\lambda$ is inaccurate as class posterior estimation, which motivates us to reannotate these augmented samples with enhanced soft labels. Aiming for precise $\lambda$ annotation, we assume that there exists a classification-optimal visual encoder $\mathrm{E}(\cdot)$ on the dataset and our semantic-mixed augmented samples. In this case, the *confidence ratio* of the two mixing classes $i$ and $j$ for the same sample $\widetilde{x}$ should be

$$\exp\left(\frac{\mathrm{E}(\widetilde{x})^\top P_i - \mathrm{E}(\widetilde{x})^\top P_j}{\tau}\right) = \frac{\lambda}{1 - \lambda},$$
$$\Leftrightarrow \lambda = \sigma\left(\frac{\mathrm{E}(\widetilde{x})^\top P_i - \mathrm{E}(\widetilde{x})^\top P_j}{\tau}\right), \quad (7)$$

where $P_k = \sum_{y_i = k} \mathrm{E}(x_i)$ is the class prototype for class $k$, $\sigma$ is the sigmoid function, $\lambda$ is the reannotated mixing coefficient replacing $\hat\lambda$, and $\tau$ is a temperature parameter. Eq. (7) is the theoretically optimal $\lambda$ annotation. However, the assumption of optimal $\mathrm{E}(\cdot)$ can be overly strong in practice. Therefore, we first introduce a feature-level $\lambda$-annotation, *i.e.* $\lambda_E$, to relax the encoded feature as

$$\mathrm{E}(\widetilde{x}) = \lambda_E P_i + (1 - \lambda_E) P_j + r,$$
$$s.t.\ r^\top (P_i - P_j) = 0, \quad (8)$$

where $\lambda_E$ serves as the interpolation coefficient and $r$ is a linearly independent residual term. Here, $\lambda_E$ is acquired by projecting $\mathrm{E}(\widetilde{x})$ onto the 1-D affine space of $P_i$ and $P_j$. Specifically, by taking the inner product of both sides of Equation (8) with $P_i - P_j$, we can express $\lambda_E$ as

$$\lambda_E = \frac{(\mathrm{E}(\widetilde{x}) - P_j)^\top (P_i - P_j)}{(P_i - P_j)^2}, \quad (9)$$

Meanwhile, Eq. (7) can be reformulated by substituting Eq. (8) into Eq. (7) as

$$\lambda = \sigma\Big(\frac{1}{\tau}\Big((\boldsymbol{P}_i - \boldsymbol{P}_j)^2(\lambda_E - \frac{1}{2}) + \frac{1}{2}(\boldsymbol{P}_i^2 - \boldsymbol{P}_j^2)\Big)\Big). \tag{10}$$

See Appendix A.1 for proof of the above Equations. It is evident that directly adopting Eq. (10) to calculate $\lambda$ would induce bias due to the unequal class norms in $\boldsymbol{P}_i^2 - \boldsymbol{P}_j^2 \neq 0$ and inter-class distances in $(\boldsymbol{P}_i - \boldsymbol{P}_j)^2 \neq (\boldsymbol{P}_k - \boldsymbol{P}_l)^2$ for $i \neq j$ and $\{i, j\} \neq \{k, l\}$, respectively. Note that $\lambda_E$ is only measured by the affine space of the class prototypes regardless of these biases. We propose to regard these factors as invariant across classes and simplify Eq. (10) as

$$\lambda = \sigma\big(s \cdot (\lambda_E - \frac{1}{2})\big), \tag{11}$$

where $s > 0$ is a scaling factor. With Eq. (11), the new annotations are free from biases inherent in the visual encoder $\mathrm{E}(\cdot)$. In practice, we adopt CLIP's visual encoder as $\mathrm{E}(\cdot)$.

**Balanced Learning**   Previous Mixup-based calibration methods adopt CE, Focal (Mukhoti et al., 2020), or certain ranking-based losses (Noh et al., 2023) for calibration with Mixup samples. We argue that they are not necessarily effective for augmented samples, especially in our case, where samples are *required* to be confusingly similar via the conditional sampling and possibly almost indistinguishable in the feature or logit space. To reveal the deficiency of common loss objectives for soft-labeled data, we define confidence-balanced loss as follows.

**Definition 3.1.**  Consider two soft-labeled data points associated with classes $i$ and $j$, with *sharper* and *softer* labels, respectively: $\widetilde{\boldsymbol{q}}_1 = \lambda_1 \boldsymbol{q}_i + (1 - \lambda_1)\boldsymbol{q}_j$ and $\widetilde{\boldsymbol{q}}_2 = \lambda_2 \boldsymbol{q}_i + (1 - \lambda_2)\boldsymbol{q}_j$, where $0.5 \leq \lambda_2 < \lambda_1 < 1$. A loss function $\mathcal{L}(\boldsymbol{p}, \boldsymbol{q}) : \Delta^K \times \Delta^K \to \mathbb{R}$ is considered a confidence-balanced loss if and only if the optimal outputs $\widetilde{\boldsymbol{p}}_1^*, \widetilde{\boldsymbol{p}}_2^*$ for the following optimization problem:

$$\min_{\widetilde{\boldsymbol{p}}_1, \widetilde{\boldsymbol{p}}_2} \mathcal{L}(\widetilde{\boldsymbol{p}}_1, \widetilde{\boldsymbol{q}}_1) + \mathcal{L}(\widetilde{\boldsymbol{p}}_2, \widetilde{\boldsymbol{q}}_2)$$
$$s.t. \ \|\widetilde{\boldsymbol{p}}_1 - \widetilde{\boldsymbol{p}}_2\|_2^2 \leq \delta \tag{12}$$

satisfy the condition that the balance function $\beta(\widetilde{\boldsymbol{p}}_1, \widetilde{\boldsymbol{p}}_2) = \|\widetilde{\boldsymbol{p}}_1 - \widetilde{\boldsymbol{q}}_1\|_2^2 - \|\widetilde{\boldsymbol{p}}_2 - \widetilde{\boldsymbol{q}}_2\|_2^2$ equals zero for all $\delta \geq 0$ and for all possible data pairs. The prediction proximity condition $\|\widetilde{\boldsymbol{p}}_1 - \widetilde{\boldsymbol{p}}_2\|_2^2 < \delta$ ensures the non-discriminability of correlated Mixup samples.

Intuitively, $\|\widetilde{\boldsymbol{p}}_1 - \widetilde{\boldsymbol{q}}_1\|_2^2$ and $\|\widetilde{\boldsymbol{p}}_2 - \widetilde{\boldsymbol{q}}_2\|_2^2$ estimate how well the model fits the two soft labels, respectively. With $\beta(\boldsymbol{p}_1, \boldsymbol{p}_2) > 0$, $\|\widetilde{\boldsymbol{p}}_2 - \widetilde{\boldsymbol{q}}_2\|_2^2$ is smaller and the model fits the softer label $\widetilde{\boldsymbol{q}}_2$ more effectively. In contrast, when $\beta(\widetilde{\boldsymbol{p}}_1, \widetilde{\boldsymbol{p}}_2) < 0$, $\|\widetilde{\boldsymbol{p}}_1 - \widetilde{\boldsymbol{q}}_1\|_2^2$ is smaller and the model adheres

more closely to the sharper label $\widetilde{\boldsymbol{q}}_1$. The network learns without such a bias only when $\beta(\widetilde{\boldsymbol{p}}_1, \widetilde{\boldsymbol{p}}_2) = 0$. These cases are also intuitively illustrated in Figure 2.

It's worth noting that the majority of common loss functions are *not* confidence-balanced. For example, ranking- or margin-based losses presume poor reliability of Mixup labels. Consequently, their learning behavior is basically blind to the exact class posteriors, therefore difficult to establish such balances. For CE and Focal losses, their patterns are both empirically and theoretically predictable. Denoting the optimal $\boldsymbol{p}_a$ as $\boldsymbol{p}_a^{CE}$ and $\boldsymbol{p}_a^{FL}$ ($a \in \{1, 2\}$) for CE and Focal losses, respectively, we can determine the sign of score $\beta$:

**Proposition 3.2.** *For $\forall \delta \geq \|\widetilde{\boldsymbol{q}}_1 - \widetilde{\boldsymbol{q}}_2\|_2^2$, we have $\beta_\delta = \beta(\boldsymbol{p}_1^{CE}, \boldsymbol{p}_2^{CE}) = 0$, while for $\forall \delta < \|\widetilde{\boldsymbol{q}}_1 - \widetilde{\boldsymbol{q}}_2\|_2^2$, we have $\beta_\delta = \beta(\boldsymbol{p}_1^{CE}, \boldsymbol{p}_2^{CE}) < 0$.*

It can be seen that similar augmented pairs can exhibit negative scores, indicating shifted confidence toward sharper class distributions. In contrast, the behavior of the Focal loss is almost the opposite in such cases, producing positive $\beta$ scores for similar samples:

**Proposition 3.3.** *For $\gamma_{FL} = 1$ and $\forall \delta < \|\widetilde{\boldsymbol{q}}_1 - \widetilde{\boldsymbol{q}}_2\|_2^2$, we have $\beta_\delta = \beta(\boldsymbol{p}_1^{FL}, \boldsymbol{p}_2^{FL}) > 0$.*

This imbalanced learning severely degrades the calibration improvement from Mixup samples by dragging the overall confidence towards certain sides. Aiming at obtaining balanced confidence, we present the following proposition:

**Proposition 3.4.** *$\forall \delta \geq 0$, we have $\beta(\boldsymbol{p}_1^{L2}, \boldsymbol{p}_2^{L2}) = 0$,*

where $\boldsymbol{p}_a^{L2}$ is the optimal solution adopting the L2 loss:

$$\mathcal{L}_{L2}(\boldsymbol{p}, \boldsymbol{q}) = \frac{1}{K}\|\boldsymbol{p} - \boldsymbol{q}\|_2^2. \tag{13}$$

The formal proofs of Propositions 3.2, 3.3, and 3.4 are provided in Appendix A.2. The proposition reveals that when two similar samples exceed the model's discriminability, $\mathcal{L}_2$ loss tends to balance the learned labels of both the harder and softer instances, instead of tending to fit a specific one of them. Notably, this objective is tailored for learning on soft-labeled data, in contrast to the calibration of one-hot-labeled samples explored in (Liang et al., 2024). In this way, we can formulate our overall objective as a simple combination of the CE and L2 losses, *i.e.*,

$$\mathcal{L}_{overall} = \sum_{i=1}^{N} \mathcal{L}_{CE}(\boldsymbol{p}_i, y_i) + \sum_{a=1}^{A} \mathcal{L}_{L2}(\widetilde{\boldsymbol{p}}_a, \widetilde{\boldsymbol{q}}_a). \tag{14}$$

# 4. Experiment

We conduct experiments with DNNs of different architectures, including ResNet-50/101 (He et al., 2016), Wide-ResNet-26-10 (Zagoruyko, 2016), and DenseNet-121

*Table 1.* **Calibration errors before and after temperature scaling.** Results in the brackets are post-temperature results. R50: ResNet-50, R101: ResNet-101. Available results on the same used settings are cited from (Noh et al., 2023).

| METHOD | CIFAR10 (KRIZHEVSKY ET AL., 2009) | | | | | | CIFAR100 (KRIZHEVSKY ET AL., 2009) | | | | | | TINY-IMAGENET (LE & YANG, 2015) | | | | | |
| | R50 (HE ET AL., 2016) | | | R101 (HE ET AL., 2016) | | | R50 (HE ET AL., 2016) | | | R101 (HE ET AL., 2016) | | | R50 (HE ET AL., 2016) | | | R101 (HE ET AL., 2016) | | |
| | ACC↑ | ECE↓ | AECE↓ | ACC↑ | ECE↓ | AECE↓ | ACC↑ | ECE↓ | AECE↓ | ACC↑ | ECE↓ | AECE↓ | ACC↑ | ECE↓ | AECE↓ | ACC↑ | ECE↓ | AECE↓ |
|---|---|---|---|---|---|---|---|---|---|---|---|---|---|---|---|---|---|---|
| CE | 95.38 | 3.75(0.97) | 2.98(1.01) | 94.46 | 3.61(0.92) | 3.55(0.85) | 77.81 | 13.59(2.93) | 13.54(2.86) | 77.48 | 12.94(2.63) | 12.94(2.66) | 64.34 | 3.18(3.18) | 2.87(2.87) | 66.04 | 3.50(3.50) | 3.52(3.52) |
| MMCE | 95.18 | 3.88(0.97) | 3.88(1.12) | 94.99 | 3.88(1.15) | 3.88(12.9) | 77.56 | 12.72(2.83) | 12.71(2.86) | 77.82 | 13.43(3.06) | 13.42(2.80) | 64.80 | 2.03(2.03) | 1.97(1.97) | 66.44 | 3.40(3.40) | 3.38(3.38) |
| ECP | 94.75 | 4.01(1.06) | 3.99(1.53) | 93.97 | 4.41(1.72) | 4.40(1.70) | 76.20 | 13.43(2.92) | 12.28(2.22) | 76.81 | 13.43(2.92) | 13.42(3.04) | 64.88 | 1.94(1.94) | 1.95(1.95) | 66.20 | 2.72(2.72) | 2.70(2.70) |
| LS | 94.87 | 3.27(1.58) | 3.67(3.02) | 94.18 | 3.35(1.51) | 3.85(3.10) | 76.45 | 6.73(4.23) | 6.54(4.26) | 76.91 | 7.99(4.38) | 7.87(4.55) | 65.46 | 3.21(2.51) | 3.23(2.51) | 65.52 | 3.11(2.51) | 2.92(2.72) |
| FL | 94.82 | 3.42(1.07) | 3.41(0.87) | 93.59 | 3.27(1.12) | 3.23(1.37) | 76.41 | 2.83(1.66) | 2.88(1.73) | 76.12 | 3.10(2.58) | 3.22(2.51) | 63.08 | 2.03(2.03) | 1.94(1.94) | 64.02 | 2.18(2.18) | 2.09(2.09) |
| MIXUP | 94.76 | 2.86(1.37) | 2.81(2.00) | 95.50 | 6.87(1.18) | 6.79(2.33) | 78.47 | 8.68(2.14) | 8.68(2.19) | 78.74 | 8.92(3.69) | 8.91(3.65) | 65.81 | 1.92(1.92) | 1.96(1.96) | 66.41 | 2.41(1.97) | 2.43(1.95) |
| FLSD | 94.77 | 3.86(0.83) | 3.74(0.96) | 93.26 | 3.92(0.93) | 3.67(0.94) | 76.20 | 2.86(2.86) | 2.86(2.86) | 76.61 | 3.29(2.04) | 3.25(1.78) | 63.56 | 1.93(1.93) | 1.98(1.98) | 64.02 | 1.85(1.85) | 1.81(1.81) |
| CRL | 95.08 | 3.14(0.96) | 3.11(1.25) | 95.04 | 3.74(1.12) | 3.73(2.03) | 77.85 | 6.30(3.43) | 6.26(3.56) | 77.60 | 7.29(3.32) | 7.14(3.31) | 64.88 | 1.65(2.35) | 1.52(2.34) | 65.87 | 3.57(1.60) | 3.56(1.52) |
| CPC | 95.04 | 5.05(1.89) | 5.04(2.60) | 95.36 | 4.78(1.52) | 4.77(2.37) | 77.23 | 13.29(3.74) | 13.28(3.82) | 77.50 | 13.32(2.96) | 13.28(3.23) | 65.70 | 3.41(3.41) | 3.42(3.42) | 66.44 | 3.93(3.93) | 3.74(3.74) |
| MBLS | 95.25 | 1.16(1.16) | 3.18(3.18) | 95.13 | 1.38(1.38) | 3.25(3.25) | 77.92 | 4.01(4.01) | 4.14(4.14) | 77.45 | 5.49(5.49) | 6.52(6.52) | 64.74 | 1.64(1.64) | 1.73(1.73) | 65.81 | 1.62(1.62) | 1.68(1.68) |
| REGMIXUP | 94.68 | 2.76(0.98) | 2.67(0.92) | 95.03 | 4.75(0.92) | 4.74(0.94) | 76.76 | 5.50(1.98) | 5.48(1.98) | 76.93 | 4.20(1.36) | 4.15(1.92) | 63.58 | 3.04(1.89) | 3.04(1.81) | 63.26 | 3.35(1.86) | 3.32(1.68) |
| AUGMIX | 94.57 | 3.29(0.63) | 3.26(0.71) | 95.02 | 3.33(0.57) | 3.32(0.81) | 77.87 | 10.93(2.49) | 10.89(2.27) | 78.63 | 11.66(2.06) | 11.66(1.85) | 65.56 | 2.64(2.10) | 2.37(2.21) | 65.89 | 2.73(2.44) | 2.78(2.47) |
| FCL | 95.44 | 0.76(0.76) | 0.75(0.75) | 95.25 | 0.95(0.95) | 1.25(1.25) | 78.19 | 3.71(2.14) | 3.74(2.11) | 78.98 | 4.19(3.23) | 4.88(3.27) | 62.67 | 7.51(1.51) | 7.51(1.45) | 64.04 | 6.97(1.70) | 6.97(1.94) |
| RANKMIXUP | 94.88 | 2.59(0.57) | 2.58(0.52) | 94.25 | 3.24(0.65) | 3.21(0.56) | 77.11 | 3.46(1.49) | 3.45(**1.42**) | 76.46 | 3.49(**1.10**) | 3.49(1.40) | 64.97 | 1.49(1.49) | 1.44(1.44) | 64.89 | 1.57(1.57) | 1.94(1.94) |
| CSM | **95.79** | **0.54(0.54)** | **0.33(0.33)** | **95.80** | **0.54(0.54)** | **0.33(0.33)** | **78.84** | **1.29(1.29)** | 1.63(1.63) | **79.17** | **1.46**(1.46) | **1.28(1.28)** | **66.99** | **1.29(1.29)** | **1.19(1.19)** | **68.20** | **1.33(1.33)** | 1.42(1.42) |

*Table 2.* **Comparison on the EQ-DATA setting** on CIFAR-10 with methods that use non-augmented data. Results in parentheses are post-temperature results.

| METRIC | LS | FL | FLSD | MBLS | FCL | CSM (OURS) |
|---|---|---|---|---|---|---|
| ACC↑ | 94.87 | 94.82 | 94.77 | 95.25 | **95.44** | 95.02 |
| ECE↓ | 3.27(1.58) | 3.42(1.07) | 3.86(0.83) | 1.16(1.16) | **0.76**(0.76) | 0.91(**0.41**) |
| AECE↓ | 3.67(3.02) | 3.41(0.87) | 3.74(0.96) | 3.18(3.18) | 0.75(0.75) | **0.68(0.42)** |

(Huang et al., 2017). We adopt CIFAR-10, CIFAR-100 (Krizhevsky et al., 2009), and Tiny-ImageNet (Le & Yang, 2015) for calibration performance and out-of-distribution (OOD) robustness comparisons. For augmentation, we adopt the class-conditional elucidating diffusion model (EDM) (Karras et al., 2022b) to generate soft-labeled samples. We select the standard ECE (Guo et al., 2017) and AECE (Ding et al., 2020) protocols as our primary evaluation metrics for calibration, while following the widely-used area under the receiver operating characteristics (AUROC) protocol (Liu et al., 2020; Pinto et al., 2022) to evaluate the OOD detection performance. Detailed descriptions of our benchmark datasets, evaluation protocols, and compared methods are presented in Appendix B.

**Training Time**   During training, data-aware calibration methods supplement each example with additional augmented samples. To control the training time, we limit the maximum number of auxiliary data associated with each dataset instance as $N_{\text{aug}} = 3$ considering the settings in (Noh et al., 2023). In fact, our method adopts $N_{\text{aug}} = 2$ for CIFAR-10/100 and $N_{\text{aug}} = 1$ for Tiny-ImageNet, displaying strong data efficiency. To make a fully fair comparison with non-augmentation methods, we also show results with a different data setting EQ-DATA: During training, we limit the number of iterations per epoch to be $\lceil \frac{\text{dataset\_size}}{(1+N_{\text{aug}})(\text{batch\_size})} \rceil$, which ensures the total amount of training data per epoch to be consistent across methods.

**Training Details**   We use our proposed method to generate augmented samples for each dataset based on the code and checkpoints from (Karras et al., 2022a) and (Wang et al., 2023b). We generate 159, 840; 158, 400; and 318, 400 CSM-augmented samples for CIFAR-10, CIFAR-100, and Tiny-ImageNet, respectively. The setup for training and testing in our work follows (Noh et al., 2023), while we also implement our approach based on their public code. Specifically, we conduct 200-epoch training on CIFAR-10 and CIFAR-100 and 100-epoch training on Tiny-ImageNet using the SGD optimizer of momentum set to 0.9. We adopt multi-step learning rate schedule which decreases from 0.1 to 0.01 and 0.001 at epochs 81 and 121 for CIFAR-10/100, or epochs 40 and 60 for Tiny-ImageNet, respectively. The weight decay is set to $5 \times 10^{-4}$. We select the scaling hyper-parameter $s$ as 4.0 for CIFAR-10/Tiny-ImageNet and 2.3 for CIFAR-100 using their validation sets.

### 4.1. Results

**State-of-the-Art Comparison**   We provide comprehensive comparisons with state-of-the-art approaches in Table 1. From the results, we can derive the following key observations: (1) Our CSM outperforms traditional regularization-based methods (*e.g.*, Label Smoothing and FLSD) by large margins via confidence-aware data augmentation, indicating the strong effectiveness of our method to leverage confidence knowledge embedded in the learning samples. (2) CSM gives comparable or superior results for ECE and ACEC consistently across all compared benchmarks including CIFAR-10, CIFAR-100, and Tiny-ImageNet, especially for the pre-temperature-scaling results. Such performance suggests that the confidence learning driven by data is crucial for model calibration, and our calibration-aware augmentation scheme can reliably generate the required sample-label pairs. (3) Our method also outperforms Mixup-based calibration methods, including the state-of-the-art RankMixup, suggesting that our generated augmentations enable better sample-confidence alignment compared to traditional Mixup outputs in terms of both fidelity and coherence. (4) With the post-hoc temperature scaling (Guo et al., 2017), optimal temperatures in our method are mostly determined as 1.0 while compared methods generally exhibit larger $T$ values, further validating that CSM effectively

*Table 3.* **Comparison on ImageNet using ResNet-50 architecture.**

| METRICS | CE | MIXUP | MBLS | REGMIXUP | RANKMIXUP | CALS | CSM (OURS) |
|---|---|---|---|---|---|---|---|
| ACC | 73.96 | 75.84 | 75.39 | 75.64 | 74.86 | 76.44 | **79.87** |
| ECE↓ | 9.10 | 7.07 | 4.07 | 5.34 | 3.93 | 1.46 | **1.32** |
| AECE↓ | 9.24 | 7.09 | 4.14 | 5.42 | 3.92 | **1.32** | 1.35 |

*Table 4.* **Comparison on ImageNet using Swin-Transformer-V2 architecture.**

| METRICS | CE | LS | FL | FLSD | MBLS | CALS | CSM (OURS) |
|---|---|---|---|---|---|---|---|
| ACC | 75.60 | 75.42 | 75.60 | 74.70 | 77.18 | 77.10 | **81.08** |
| ECE↓ | 9.95 | 7.32 | 3.19 | 2.44 | 1.95 | 1.61 | **1.49** |
| AECE↓ | 9.94 | 7.33 | 3.18 | 2.37 | 1.73 | **1.69** | 1.86 |

solves the over-confidence issue embedded in ordinary one-hot datasets and the compared learning paradigms.

We also share results on larger datasets or architectures by comparing CSM with representative methods on ImageNet with the ResNet-50 and Swin-Transformer architectures in Table 3 and Table 4, respectively. Our method performs equally or more effectively compared to these methods, especially to the mixup-based methods.

We also provide qualitative comparisons using Wide-ResNet-26-10 (Zagoruyko, 2016) and DenseNet-121 (Huang et al., 2017) as the network in Appendix C, which demonstrate similar properties to those in Table 1. Meanwhile, we provide results on the EQ-DATA setting in Table 2 for comparisons with non-augmentation methods for equalized training time. It is observed that our CSM can still achieve competitive or superior performance, particularly in consistently superior post-temperature ECE and AECE. Comparisons of the estimated training time is provided in Appendix C, where one can observe that the number of augmented samples per batch is the major factor affecting training time.

**Out-of-Distribution Detection**   To validate the network calibration from different perspectives, we conduct qualitative comparisons for the out-of-distribution (OOD) detection with the proposed CSM. We measure the sample uncertainty by the entropy of the softmax output $p_i = \text{softmax}(f_\theta(x_i))$. Therefore, we report AUROC scores comparisons in Table 5. In the table, our proposed CSM achieves superior and comparable results on the CIFAR-10 and CIFAR-100 datasets, respectively. Such results indicate that our approach learns precise sample confidence levels and is therefore also superior in detecting distributional shifts, verifying the significance of explicit soft-labeled samples in OOD detection.

### 4.2. Discussions

We discuss the properties of our method with extensive experiments below. More results are available in Appendix C.

*Table 5.* **AUROC (%) for robustness evaluation under distribution shifts.** Higher values indicate better performance. We use CIFAR-10 and CIFAR-100 as in-distribution datasets, with each serving as the OOD dataset for the other, alongside an external Tiny-ImageNet dataset.

| ID DATASET | CIFAR-10 | | CIFAR-100 | |
|---|---|---|---|---|
| OOD DATASET | CIFAR-100 | TINY-IMAGENET | CIFAR-10 | TINY-IMAGENET |
| TS | 86.73 | 88.06 | 76.38 | 80.12 |
| LS | 75.23 | 76.29 | 72.92 | 77.54 |
| FL | 85.63 | 86.84 | 78.37 | 81.23 |
| MIXUP | 82.07 | 84.09 | 78.46 | **81.55** |
| CPC | 85.28 | 85.15 | 74.68 | 76.66 |
| MBLS | 86.20 | 87.55 | 73.88 | 79.19 |
| REGMIXUP | 87.82 | 87.54 | 76.00 | 80.72 |
| RANKMIXUP | 87.82 | 88.94 | 78.64 | 80.67 |
| CSM | **91.82** | **91.05** | **79.02** | 80.45 |

*Table 6.* **Ablation Study.** Variants regarding augmentations, annotations, and learning objectives are compared. Results are from CIFAR-100 with ResNet-50.

| VARIANTS | ACC↑ | ECE ↓ | AECE ↓ | ECE$^{PT}$ ↓ | ECE$^{PT}$ ↓ | $T$ |
|---|---|---|---|---|---|---|
| 1. W/O AUGMENTED DATA | 77.48 | 12.94 | 12.94 | 2.63 | 2.66 | 1.6 |
| 2. CSM (L2) W/O REANNO. | 78.92 | 5.20 | 5.20 | 3.33 | 3.24 | 1.2 |
| 3. CSM (L2) W/ CLIP LABELS. | 66.60 | 52.78 | 52.78 | - | - | - |
| 4. CSM (L2) W/ ONE-HOT AUG. | **79.24** | 10.84 | 10.84 | 2.48 | 2.41 | 1.5 |
| 5. MIXUP (CE) | 78.47 | 8.68 | 8.68 | 2.14 | 2.19 | 1.3 |
| 6. MIXUP (FL) | 79.64 | 2.45 | 2.44 | 2.45 | 2.44 | 1.0 |
| 7. MIXUP (L2) | 78.88 | 2.49 | 2.62 | 2.49 | 2.62 | 1.0 |
| 8. CSM (CE) | 78.93 | 2.47 | 2.14 | 2.47 | 2.14 | 1.0 |
| 9. CSM (FL) | 77.65 | 2.43 | 2.30 | 2.16 | 2.09 | 0.9 |
| 10. CSM (L2, OURS) | 78.84 | **1.29** | **1.63** | **1.29** | **1.63** | 1.0 |

**Ablation Study**   We compare our CSM with variants in Table 6, testing different augmentation types, soft labels, and loss functions. We can observe that Mixup and CSM variants outperform the standard CE loss, improving both ECE and accuracy, highlighting the importance of data-driven network calibration. Compared with the variant w/o reannotation (or using the generation labels, Var. 2), our reannotated variant produces significantly better calibration performances, validating the accurate confidence estimation in our refined labels. Meanwhile, using the vanilla CLIP annotations (Var. 3) yields the worst ACC and pre-temperature calibration errors, primarily due to the noisy information by annotating all classes. Directly adopting class-conditioned augmentations from the diffusion model (Var. 4) can slightly rise the prediction accuracy, as also evidenced by the augmented classification literature. However, it fails to improve model calibration due to the lack of soft-labeled samples.

Among the losses used in CSM (Var. 8-10), the L2 loss is the only one that reduces ECE, AECE, and post-TS results below 2.0, suggesting it effectively balances learned confidence in augmentations. In contrast, CE and FL losses often require temperature adjustments (Var. 5,9), with CE favoring sharper labels and FL for softer ones, aligning with our theoretical expectations from Section 3.3. CSM variants, particularly with CE and L2 losses, significantly outperform traditional Mixup in terms of calibration, verifying the effectiveness of CSM's realistic in-domain augmentations.

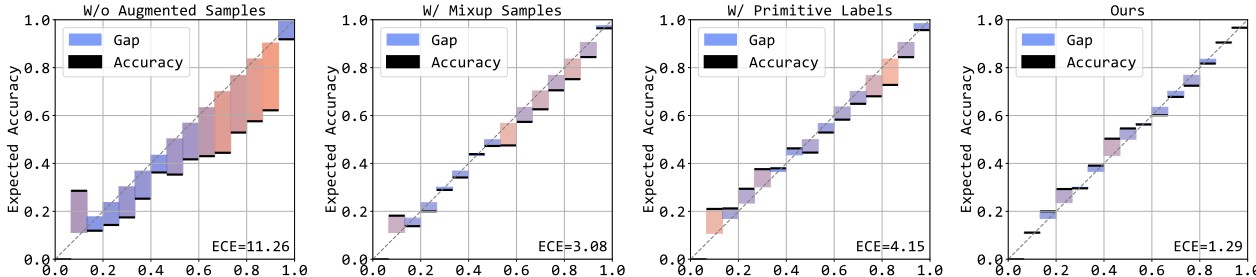

*Figure 3.* **Reliability diagram** of various methods before temperature scaling.

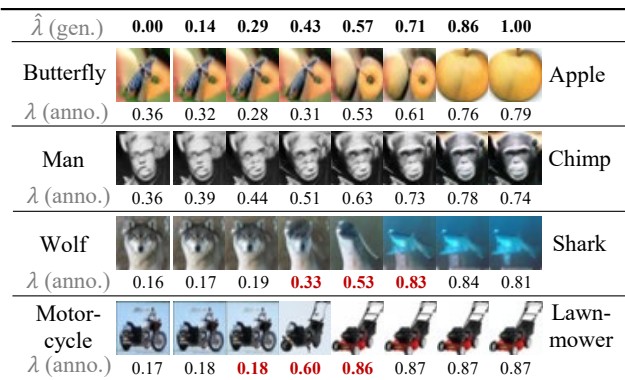

*Figure 4.* **Visualization of generated samples**, including augmented sample sets, mixing coefficient $\hat{\lambda}$ during generation (gen.), and their reannotated $\lambda$ (anno.) for soft labels.

However, applying L2 loss to the traditional Mixup samples does not yield significantly better results (Var. 5-7). It is noteworthy that CSM uses conditioned dual-sampling of one-hot samples $\widetilde{x}^0$ and $\widetilde{x}^1$, which are correlated considering joint latent variable $x^{(T)}$, while every pair of them in Mixup is sampled independently, which makes the soft-labeled samples much easier to be distinguished apart by the model. Therefore, traditional Mixup with L2 loss cannot perform noticeably better on all metrics.

**Augmented Samples and Annotations**   To analyze the properties of augmented samples, we visualize some case images in Fig. 4. It can be seen that CSM can produce semantically sound intermediate samples compared to Mixup's simple overlap. Meanwhile, these samples exhibit characteristics of both classes with gradual semantic change from one category to another. For images with abrupt transitions (in red numbers) or inaccurate class posteriors, our annotation paradigm yields more precise $\lambda$ values compared to the generation phase $\hat{\lambda}$.

**Confidence Value Adjustment**   To analyze the learned

confidence values, we compare the reliability diagrams of different methods across Figure 3. We examine CSM alongside three variants: 1) without augmentations, 2) with vanilla Mixup, and 3) using labels from generation. The augmentation-free variant shows severe over-confidence across the middle-to-high confidence range, while the Mixup variant also exhibits over-confidence, though less severely. Regularization-based methods that calibrate with one-hot labels can show similar over-confidence, highlighting common issues in confidence estimation. CSM with primitive labels $\hat{\lambda}$, however, is both over- and under-confident at higher and lower ranges, respectively, suggesting a misalignment between annotations and confidence levels. In contrast, CSM demonstrates precise confidence estimation across both low and high ranges, with only slight errors in the middle, confirming the effectiveness of our confidence-aware augmentation.

We also examine the distribution of top confidence values in Figure 5(c). Without soft labels, predicted confidence peaks at 1.00, while Mixup-augmented training peaks at around 0.99. Our CSM produces a more even distribution, effectively reducing the over-confidence observed in calibration methods using one-hot labels.

**Calibration through Training**   To dynamically study the training-time behavior of CSM, we plot the Negative Log-Likelihood (NLL) and Overconfidence Errors (OE) in Figure 5(a)-(b) using exponential moving averages for better visualization. From the plot, we can observe that better classification has already been achieved in early epochs while the strong calibration is not evident until the second adjustment of the learning rate. This is mainly due to the order of network fitting, which prioritizes easy one-hot samples but learns the actual confidence in hard soft labels later. The lower calibration error of Mixup during early epochs also indicates that the larger difficulty in fitting confidence of our augmentations compared to Mixup samples. However, CSM can eventually achieve lower calibration error with a fine-grained learning rate for better fitting, validating the effectiveness of our proposed learning scheme.

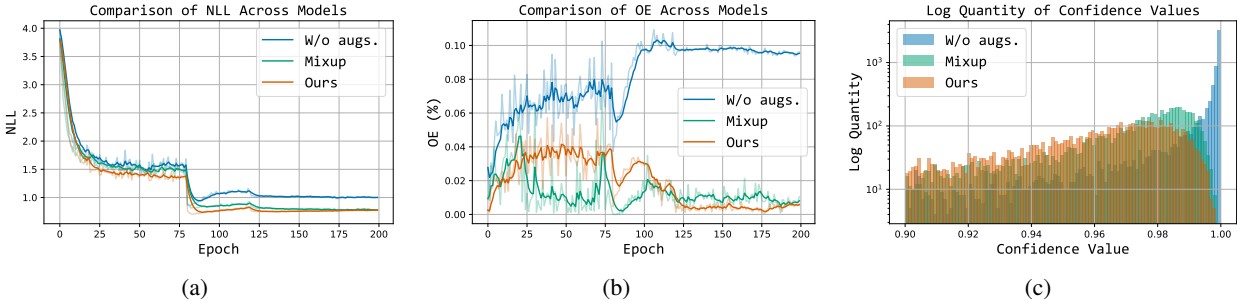

(a)  (b)  (c)

*Figure 5.* **Training Characteristics.** (a) NLL values of different methods on CIFAR-100 validation split across training epochs. (b) Over-confidence Errors (OE) of different methods on CIFAR-100 validation split across training epochs. (c) Log-quantity of top confidence values in CIFAR-100 test split among different methods.

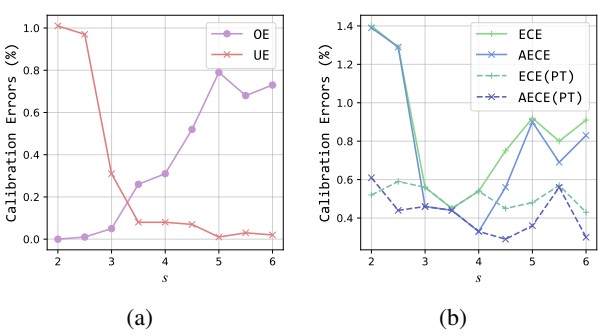

(a)  (b)

*Figure 6.* **Analysis of the hyperparameter** $s \in [2.0, 6.0]$. (a) OE and UE *w.r.t.* different $s$ values. (b) ECE/AECE and their post-temperature values *w.r.t* different $s$ values.

**Scaling Factor** To analyze the influence of hyperparameter $s$, we plot various calibration results on CIFAR-10 in Figure 6, including Overconfidence Errors (OE) and Underconfidence Errors (UE) (Thulasidasan et al., 2019). Due to the scaling effect of $s$ values to the confidence levels, the label distribution can shift towards shaper and softer ones with higher and lower $s$, respectively. This is evident in Figure 6(a), where OE increases while UE decreases when the labels are shifted towards traditional one-hot annotations by large $s$, meaning that the model behaves over-confident about predictions. Consequently, ECE and AECE values increase a lot when $s$ leaves a certain range as shown in Figure 6(b). Nevertheless, with post-hoc temperature scaling, the calibration errors can become consistently low regardless of $s$ values, verifying the stability of the proposed CSM.

**Integrating Test-time Methods** While CSM introduces training-time augmentations for softened labels, there exists test-time augmentation methods targeting the same goal. Meanwhile, proximity problems have also been investigated in inference time. We conduct experiments to combine these methods with our CSM, as demonstrated in Table 7. Inte-

*Table 7.* **Integration with Test-Time Calibration Methods.**

| VARIANTS | ECE↓ | AECE ↓ | PIECE($Xiong et al.$, 2023) ↓ |
|---|---|---|---|
| CSM | **1.29** | 1.63 | 3.16 |
| CSM + TTA | 1.39 | **1.53** | 3.15 |
| CSM + PROCAL | 1.89 | 1.82 | **3.11** |

grated with test-time augmentations (TTA), the method balances ECE and AECE effectively, achieving an optimized AECE of 1.53 on CIFAR-100. Compared to (Hekler et al., 2023) using test-time sample-wise scaling, CSM employs training-time augmentation with inter-sample augmentations to expand the proximity space, enhancing calibration robustness. Combined with proximity method ProCal, we find that the proximity-informed metric PIECE displays better results, which validates the robustness growth related to proximity from the integration. Despite these growths, the overall calibration improvement is relatively small, indicating the effective sample augmentation and debiased proximity learning of our method.

## 5. Conclusion

In conclusion, CSM is a novel framework designed to bridge the optimization gap between the calibration objective and the data of full certainty. By generating augmented samples with semantic mixing and reannotating them with confidence scores via diffusion models, CSM enables a more accurate alignment between model predictions and their true likelihoods. Our exploration of balanced loss functions further enhances the new data representation paradigm, enhancing it an integrated pipeline for superior model calibration. Theoretical and practical evidence validate the effectiveness of CSM for strong sample-label augmentations. Meanwhile, various ablation results demonstrate CSM's balanced learning of true confidence levels. The framework's ability to acquire meaningful augmentation positions it as a novel baseline for semantic-aware confidence calibration.

## Acknowledgements

This work was supported in part by the Start-up Grant (No. 9610680) of the City University of Hong Kong, Young Scientist Fund (No. 62406265) of NSFC, and the Australian Research Council under Projects DP240101848 and FT230100549.

## Impact Statement

This paper presents work whose goal is to advance the field of Machine Learning. There are many potential societal consequences of our work, none which we feel must be specifically highlighted here.

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

# Appendix

## A. Proofs of Equations and Propositions

### A.1. Proofs of Equation (10)

As we have assumed in advance that a fully optimized model $\mathrm{E}(\cdot)$ should satisfy

$$\lambda = \sigma\big(\frac{\mathrm{E}(\widetilde{\boldsymbol{x}})^{\top}\boldsymbol{P}_i - \mathrm{E}(\widetilde{\boldsymbol{x}})^{\top}\boldsymbol{P}_j}{\tau}\big), \tag{15}$$

and we can relax the encoded feature and acquire a feature-level mixing coefficient $\lambda_E$ through

$$\mathrm{E}(\widetilde{\boldsymbol{x}}) = \lambda_E \boldsymbol{P}_i + (1 - \lambda_E)\boldsymbol{P}_j + \boldsymbol{r},$$
$$s.t. \ \boldsymbol{r}^{\top}(\boldsymbol{P}_i - \boldsymbol{P}_j) = 0, \tag{16}$$

we can first rewrite Equation (16) by vector multiplying $(\boldsymbol{P}_i - \boldsymbol{P}_j)$ on both sides as

$$\mathrm{E}^{\top}(\widetilde{\boldsymbol{x}})(\boldsymbol{P}_i - \boldsymbol{P}_j) = \lambda_E \boldsymbol{P}_i^{\top}(\boldsymbol{P}_i - \boldsymbol{P}_j) + (1 - \lambda_E)\boldsymbol{P}_j^{\top}(\boldsymbol{P}_i - \boldsymbol{P}_j) + \boldsymbol{r}^{\top}(\boldsymbol{P}_i - \boldsymbol{P}_j), \tag{17}$$

which can be reformulated as

$$(\boldsymbol{P}_i - \boldsymbol{P}_j)^2 \lambda_E = (\mathrm{E}(\widetilde{\boldsymbol{x}}) - \boldsymbol{P}_j)^{\top}(\boldsymbol{P}_i - \boldsymbol{P}_j) - \boldsymbol{r}^{\top}(\boldsymbol{P}_i - \boldsymbol{P}_j)$$
$$= (\mathrm{E}(\widetilde{\boldsymbol{x}}) - \boldsymbol{P}_j)^{\top}(\boldsymbol{P}_i - \boldsymbol{P}_j) \tag{18}$$

Therefore, $\lambda_E$ can be expressed as

$$\lambda_E = \frac{(\mathrm{E}(\widetilde{\boldsymbol{x}}) - \boldsymbol{P}_j)^{\top}(\boldsymbol{P}_i - \boldsymbol{P}_j)}{(\boldsymbol{P}_i - \boldsymbol{P}_j)^2}, \tag{19}$$

which is independent of $\boldsymbol{r}$.

Meanwhile, substituting Equation (16) into Equation (15), we can acquire

$$\begin{aligned}\lambda &= \sigma\Big(\frac{1}{\tau}\Big(\lambda_E \boldsymbol{P}_i^{\top}(\boldsymbol{P}_i - \boldsymbol{P}_j) + (1 - \lambda_E)\boldsymbol{P}_j^{\top}(\boldsymbol{P}_i - \boldsymbol{P}_j) + \boldsymbol{r}^{\top}(\boldsymbol{P}_i - \boldsymbol{P}_j)\Big)\Big)\\ &= \sigma\Big(\frac{1}{\tau}\Big((\boldsymbol{P}_i - \boldsymbol{P}_j)^2 \lambda_E + \boldsymbol{P}_j^{\top}(\boldsymbol{P}_i - \boldsymbol{P}_j)\Big)\Big)\\ &= \sigma\Big(\frac{1}{\tau}\Big((\boldsymbol{P}_i - \boldsymbol{P}_j)^2(\lambda_E - \frac{1}{2}) + \frac{1}{2}(\boldsymbol{P}_i - \boldsymbol{P}_j)^2 + \boldsymbol{P}_j^{\top}(\boldsymbol{P}_i - \boldsymbol{P}_j)\Big)\Big)\\ &= \sigma\Big(\frac{1}{\tau}\Big((\boldsymbol{P}_i - \boldsymbol{P}_j)^2(\lambda_E - \frac{1}{2}) + \frac{1}{2}\boldsymbol{P}_i^2 + \frac{1}{2}\boldsymbol{P}_j^2 - \boldsymbol{P}_i^{\top}\boldsymbol{P}_j + \boldsymbol{P}_i^{\top}\boldsymbol{P}_j - \boldsymbol{P}_j^2\Big)\Big)\\ &= \sigma\Big(\frac{1}{\tau}\Big((\boldsymbol{P}_i - \boldsymbol{P}_j)^2(\lambda_E - \frac{1}{2}) + \frac{1}{2}(\boldsymbol{P}_i^2 - \boldsymbol{P}_j^2)\Big)\Big).\end{aligned} \tag{20}$$

Therefore, $\lambda_E$ and $\lambda$ can be expressed with Equation (19) and Equation (20), respectively.

### A.2. Proofs of Proposition 3.2, Proposition 3.3, and Proposition 3.4.

We start our proofs through restating the definition of a mixup-balanced Loss without the tilde hat for simplicity.

**Mixup-balanced Loss.** We regard a loss function $\mathcal{L}(\boldsymbol{p}, \boldsymbol{q}) : P^K \times P^K \to \mathbb{R}$ as mixup-balanced loss iff the optimal $\boldsymbol{p}_1^*, \boldsymbol{p}_2^*$ for

$$\min_{\boldsymbol{p}_1, \boldsymbol{p}_2} \mathcal{L}(\boldsymbol{p}_1, \boldsymbol{q}_1) + \mathcal{L}(\boldsymbol{p}_2, \boldsymbol{q}_2)$$
$$s.t. \ \|\boldsymbol{p}_1 - \boldsymbol{p}_2\|_2^2 \le \delta \tag{21}$$

is a root for score function $\beta(\boldsymbol{p}_1, \boldsymbol{p}_2) = \|\boldsymbol{p}_1 - \boldsymbol{q}_1\|_2^2 - \|\boldsymbol{p}_2 - \boldsymbol{q}_2\|_2^2$ for all $\delta \geq 0$, where $P^K$ is the K-dimensional probability simplex space. $\boldsymbol{q}_i$ $(i = 1, 2)$ is the annotated class probability for mixup or perturbed samples of the same source image but with different shifting strengths, *i.e.*,

$$\begin{cases} \boldsymbol{x}_i = \alpha_i \boldsymbol{x} + (1 - \alpha_i)\boldsymbol{x}', \\ \boldsymbol{q}_i = \alpha_i \boldsymbol{l}_s + (1 - \alpha_i)\boldsymbol{l}_t, \end{cases}$$
$$0.5 \leq \alpha_i < 1, i = 1, 2, \tag{22}$$

where $\boldsymbol{l}_s$ and $\boldsymbol{l}_t$ are one-hot labels of the source and target classes $s$ and $t$, respectively. The above prediction proximity condition $\|\boldsymbol{p}_1 - \boldsymbol{p}_2\|_2^2 < \delta$ is imposed to outline the non-discriminability of correlated mixup samples. Without losing generality, we always assume $0.5 \leq \alpha_2 < \alpha_1 < 1$ in the following proofs.

## A.3. Proof of Proposition 3.4

The L2 loss is defined as $\mathcal{L}_{L2}(\boldsymbol{p}, \boldsymbol{q}) = \|\boldsymbol{p} - \boldsymbol{q}\|_2^2$, which is mixup-balanced. To prove it, we need to solve

$$\min_{\boldsymbol{p}_1, \boldsymbol{p}_2} \|\boldsymbol{p}_1 - \boldsymbol{q}_1\|_2^2 + \|\boldsymbol{p}_2 - \boldsymbol{q}_2\|_2^2$$
$$s.t. \ \|\boldsymbol{p}_1 - \boldsymbol{p}_2\|_2^2 \leq \delta. \tag{23}$$

a) When $\delta \geq \|\boldsymbol{q}_1 - \boldsymbol{q}_2\|_2^2$, the optimal solution $\boldsymbol{p}_1^* = \boldsymbol{q}_1$ and $\boldsymbol{p}_2^* = \boldsymbol{q}_2$ is mixup-balanced.

b) When $\delta < \|\boldsymbol{q}_1 - \boldsymbol{q}_2\|_2^2$, we represent the optima as

$$\begin{cases} \boldsymbol{p}_1^* = \lambda_1 \boldsymbol{q}_1 + (1 - \lambda_1)\boldsymbol{q}_2 + \boldsymbol{r}_1, \\ \boldsymbol{p}_2^* = \lambda_2 \boldsymbol{q}_2 + (1 - \lambda_2)\boldsymbol{q}_1 + \boldsymbol{r}_2, \end{cases} \tag{24}$$

where $\boldsymbol{r}_1^\top(\boldsymbol{q}_1 - \boldsymbol{q}_2) = \boldsymbol{r}_2^\top(\boldsymbol{q}_1 - \boldsymbol{q}_2) = 0$. The objective in Eq. 23 is reformulated as

$$\min_{\boldsymbol{p}_1, \boldsymbol{p}_2} \|\boldsymbol{p}_1 - \boldsymbol{q}_1\|_2^2 + \|\boldsymbol{p}_2 - \boldsymbol{q}_2\|_2^2$$
$$= \|(1 - \lambda_1)(\boldsymbol{q}_2 - \boldsymbol{q}_1) + \boldsymbol{r}_1\|_2^2 + \|(1 - \lambda_2)(\boldsymbol{q}_1 - \boldsymbol{q}_2) + \boldsymbol{r}_2\|_2^2. \tag{25}$$

Temporarily, we regard $\boldsymbol{p}_i^*$ as vectors in $\mathbb{R}^K$ as we'll see the optimal ones always lie in $P^K$. With such relaxation, $\boldsymbol{r}_i$ can take any vector value in their subspace, but it optimizes Eq. (25) only when $\boldsymbol{r}_i = \boldsymbol{0}$, further simplifying Eq. (25) as $\min_{\lambda_1, \lambda_2} (1 - \lambda_1)^2 + (1 - \lambda_2)^2$. Substituting Eq. (24) into the proximity condition, we get $(\lambda_1 + \lambda_2 - 1)^2 \leq \frac{\delta}{\|\boldsymbol{q}_1 - \boldsymbol{q}_2\|_2^2}$, *i.e.*, $1 - \sqrt{\frac{\delta}{\|\boldsymbol{q}_1 - \boldsymbol{q}_2\|_2^2}} \leq \lambda_1 + \lambda_2 \leq 1 + \sqrt{\frac{\delta}{\|\boldsymbol{q}_1 - \boldsymbol{q}_2\|_2^2}}$. For each valid value of $\lambda_1 + \lambda_2 = A$, we consider Lagrangian equation

$$L(\lambda_1, \lambda_2, \beta) = (1 - \lambda_1)^2 + (1 - \lambda_2)^2 + \beta(\lambda_1 + \lambda_2 - A), \tag{26}$$

where $\beta$ is the Lagrangian multiplier. By setting the derivative *w.r.t.* $\lambda_i$ to 0, we solve for the optimum $\lambda_i^*$ as

$$\left. \frac{\partial L(\lambda_1, \lambda_2, \beta)}{\partial \lambda_i} \right|_{\lambda_i = \lambda_i^*} = 2(\lambda_i^* - 1) + \beta = 0$$
$$2(1 - \lambda_1^*) = 2(1 - \lambda_2^*) = \beta$$
$$\lambda_1^* = \lambda_2^* = \frac{A}{2} \in (0, 1). \tag{27}$$

Therefore, $\boldsymbol{p}_i^*$s are actually interpolations of $\boldsymbol{q}_i$s which definitely lie in $P^K$. We have $\|\boldsymbol{p}_1^* - \boldsymbol{q}_1\|_2^2 = \|\boldsymbol{p}_2^* - \boldsymbol{q}_2\|_2^2 = (1 - \frac{A}{2})\|\boldsymbol{q}_1 - \boldsymbol{q}_2\|_2^2$ by substituting Eq. (27) and (24) into each term, indicating $\forall A, \beta(\boldsymbol{p}_1^*, \boldsymbol{p}_2^*) = 0$. Hence, the L2 loss is mixup-balanced.

## A.4. Proofs of Proposition 3.2 and Proposition 3.3

CE and Focal Losses *are not* Mixup-balanced. We prove that $\beta \leq 0$ is always true for the cross entropy loss while for Focal loss with some specific $\gamma$, $\beta \geq 0$ is always the case.

We denote a dual-sample loss function as $\mathcal{L}^{Pair} = \sum_{i=1}^{2} \sum_{k=1}^{K} q_i^k \mathcal{L}_{Item}(p_i^k)$, where $\mathcal{L}_{Item}(\cdot)$ is monotonous and approaches infinity when $\boldsymbol{p}_i^k$ approaches 0. To solve the loss minimization objective

$$\min_{\boldsymbol{p}_1, \boldsymbol{p}_2} \mathcal{L}^{Pair} \ \ s.t. \ \|\boldsymbol{p}_1 - \boldsymbol{p}_2\|_2^2 \leq \delta, \sum_k p_1^k = 1, \sum_k p_2^k = 1, \tag{28}$$

we should first clarify several points before looking into it.

a) For every class $k \neq s, t$, $q_i^k = 0$, the optimal $p_i^k$ is 0. Otherwise, we can carry out 3 types of operations without increasing $\|\boldsymbol{p}_1 - \boldsymbol{p}_2\|_2^2$. Op. 1 is formulated as

$$p_1^k \leftarrow p_1^k - r, \ \ p_2^k \leftarrow p_2^k - r,$$
$$p_1^c \leftarrow p_1^c + r, \ \ p_2^c \leftarrow p_2^c + r, \tag{29}$$

where $c \in \{s, t\}$ and $r > 0$. While this operation is proximity-invariant, it reduces the objective function and can finally decrease at least one of $\{p_i^k, p_j^k\}$ to 0 for every pair of them indexed by $k$. Then, Op. 2 is given as

$$p_1^{k_1} \leftarrow p_1^{k_1} - r, \ \ p_2^{k_2} \leftarrow p_2^{k_2} - r,$$
$$p_1^c \leftarrow p_1^c + r, \ \ p_2^c \leftarrow p_2^c + r, \tag{30}$$

where $k_1, k_2 \notin \{s, t\}$, $k_1 \neq k_2$. Op. 2 optimizes both the objective and the proximity value, making only one of $\{\boldsymbol{p}_1, \boldsymbol{p}_2\}$ (denoted as $\boldsymbol{p}_i$) having spare values. Finally, Op. 3 merges these values into $p_i^s$ or $p_i^t$:

$$p_i^k \leftarrow p_i^k - r,$$
$$p_i^c \leftarrow p_i^c + r, \tag{31}$$

where before operation, $p_i^c < p_j^c (i \neq j)$ and $r \leq p_j^c - p_i^c$. It's easy to prove the existence of such class $c$ and the reduction of both the objective and the proximity value.

With these operations, we can optimize the objective when $p_i^k \neq 0$ for any class $k \neq s, t$. Therefore, $p_i^k = 0$ is always true in the optimal $\boldsymbol{p}_i^*$, i.e., $p^*{}_i^s + p^*{}_i^t = 1$.

b) As we have assumed in advance that $q_2^s = \alpha_2 < \alpha_1 = q_1^s$, we can now immediately derive that $p^*{}_2^s \leq p^*{}_1^s$. Otherwise, we can swap $\boldsymbol{p}_1^*$ and $\boldsymbol{p}_2^*$ to further lower the loss while keeping the proximity constraint unchanged. This helps us to derive

$$\|\boldsymbol{p}_1 - \boldsymbol{p}_2\|_2^2 = (p_1^s - p_2^s)^2 + (p_1^t - p_2^t)^2$$
$$= (p_1^s - p_2^s)^2 + (1 - p_1^s - 1 + p_2^s)^2$$
$$= 2(p_1^s - p_2^s)^2 = A,$$
$$p^*{}_1^s = p^*{}_2^s + \sqrt{\frac{A}{2}} \tag{32}$$

for a specific value of $A \in [0, \delta]$. For simplicity, we rewrite Eq. (32) as:

$$p^*{}_1^s = p^*{}_2^s + \epsilon, \tag{33}$$

where $\epsilon \in \left[0, \sqrt{\frac{\delta}{2}}\right]$.

These observations help us formulate and solve the following Lagrangian equation (for arbitrary $A$):

$$L(\boldsymbol{p}_1, \boldsymbol{p}_2, \beta, \beta_1, \beta_2)$$
$$= \mathcal{L}^{Pair} + \beta(\|\boldsymbol{p}_1 - \boldsymbol{p}_2\|_2^2 - A) + \beta_1(\sum_k p_1^k - 1) + \beta_2(\sum_k p_2^k - 1), \tag{34}$$

where $\beta$, $\beta_1$ and $\beta_2$ are the Lagrangian multipliers. By taking derivatives w.r.t. $p_i^k$ and set it to 0, we have:

$$\frac{\partial L}{\partial p_i^k} = \frac{\partial \mathcal{L}^{Pair}}{\partial p_i^k} + 2\beta(p_i^k - p_j^k) + \beta_i = 0, \tag{35}$$

where $i, j \in \{1, 2\}$, $i \neq j$, and $k \in \{s, t\}$. It can be seen that

$$\frac{\partial L}{\partial p_i^s} + \frac{\partial L}{\partial p_i^t} = \frac{\partial \mathcal{L}^{Pair}}{\partial p_i^s} + \frac{\partial \mathcal{L}^{Pair}}{\partial p_i^t} + 2\beta(p_i^s - p_j^s + p_i^t - p_j^t) + 2\beta_i = 0$$

$$\frac{\partial L}{\partial p_i^s} + \frac{\partial L}{\partial p_i^t} = \frac{\partial \mathcal{L}^{Pair}}{\partial p_i^s} + \frac{\partial \mathcal{L}^{Pair}}{\partial p_i^t} + 2\beta_i = 0$$

$$\beta_i = -\frac{1}{2}\Big(\frac{\partial \mathcal{L}^{Pair}}{\partial p_i^s} + \frac{\partial \mathcal{L}^{Pair}}{\partial p_i^t}\Big), \tag{36}$$

and

$$\frac{\partial L}{\partial p_1^k} + \frac{\partial L}{\partial p_2^k} = \frac{\partial \mathcal{L}^{Pair}}{\partial p_1^k} + \frac{\partial \mathcal{L}^{Pair}}{\partial p_2^k} + 2\beta(p_1^k - p_2^k + p_2^k - p_1^k) + \beta_1 + \beta_2 = 0$$

$$\frac{\partial L}{\partial p_1^k} + \frac{\partial L}{\partial p_2^k} = \frac{\partial \mathcal{L}^{Pair}}{\partial p_1^k} + \frac{\partial \mathcal{L}^{Pair}}{\partial p_2^k} + \beta_1 + \beta_2 = 0. \tag{37}$$

By substituting Eq. (36) into Eq. (37) and let $k = s$ ($k = t$ yields the same result), we have:

$$\frac{\partial \mathcal{L}^{Pair}}{\partial p_1^s} + \frac{\partial \mathcal{L}^{Pair}}{\partial p_2^s} - \frac{1}{2}\Big(\frac{\partial \mathcal{L}^{Pair}}{\partial p_1^s} + \frac{\partial \mathcal{L}^{Pair}}{\partial p_1^t}\Big) - \frac{1}{2}\Big(\frac{\partial \mathcal{L}^{Pair}}{\partial p_2^s} + \frac{\partial \mathcal{L}^{Pair}}{\partial p_2^t}\Big) = 0.$$

$$\frac{\partial \mathcal{L}^{Pair}}{\partial p_1^s} - \frac{\partial \mathcal{L}^{Pair}}{\partial p_1^t} + \frac{\partial \mathcal{L}^{Pair}}{\partial p_2^s} - \frac{\partial \mathcal{L}^{Pair}}{\partial p_2^t} = 0. \tag{38}$$

**CE Loss.** When $\delta \geq \|q_1 - q_2\|_2^2$, the optimal solution in the case of CE loss is $p_1^* = q_1$ and $p_2^* = q_2$, which is mixup-balanced. However, when $\delta < \|q_1 - q_2\|_2^2$, the CE loss always yield negative $\beta$ scores, indicating its preference for fitting samples close to the source.

We prove this by regarding the left part of Eq. (38) as a function of $p_2^s$ since we have $p_1^{*s} = p_2^{*s} + \epsilon$ and $p_i^{*s} + p_i^{*t} = 1$. From here on, we discuss the optimal $p_i$ and omit the star superscript for simplicity. The left part of Eq. (38) can be rewritten as

$$h(x) = -\frac{\alpha_1}{x + \epsilon} + \frac{1 - \alpha_1}{1 - x - \epsilon} - \frac{\alpha_2}{x} + \frac{1 - \alpha_2}{1 - x}. \tag{39}$$

The root of $h(x)$ in $[0.5, 1 - \epsilon]$ is the root $p_2^s$ of Eq. (38), i.e., $p_2^{*s}$. Similarly, the root of $h(x - \epsilon)$ is $p_1^{*s}$. Although it is difficult to find a closed form of their roots, we can find the mean of them, i.e., $t = (p_1^{*s} + p_2^{*s})/2$, as the root of $h(x - \epsilon/2)$. Our initial goal is to find the sign of $\beta$, which is now

$$\beta = (p_1^s - \alpha_1)^2 + (1 - p_1^s - 1 + \alpha_1)^2 - (p_2^s - \alpha_2)^2 - (1 - p_2^s - 1 + \alpha_2)^2$$
$$= 2(p_1^s - \alpha_1)^2 - 2(p_2^s - \alpha_2)^2$$
$$= 2(2t - \alpha_1 - \alpha_2)(\epsilon + \alpha_2 - \alpha_1) \tag{40}$$

and is closely associated with $t$. The sign of $\beta$ is determined by the sign of $(\alpha_1 + \alpha_2)/2 - t$. It's intractable to solve $t$ directly, but noting the monotonicity of $f$ with

$$h'(x) = \frac{\alpha_1}{(x + \epsilon)^2} + \frac{1 - \alpha_1}{(1 - x - \epsilon)^2} + \frac{\alpha_2}{x^2} + \frac{1 - \alpha_2}{(1 - x)^2} > 0 \tag{41}$$

for all $x \in (0, 1 - \epsilon)$, we now have

$$\frac{\alpha_1 + \alpha_2}{2} - t < 0 \Leftrightarrow h\Big(\frac{\alpha_1 + \alpha_2}{2} - \frac{\epsilon}{2}\Big) < h\Big(t - \frac{\epsilon}{2}\Big) = 0, \tag{42}$$

where $h(\frac{\alpha_1 + \alpha_2}{2} - \frac{\epsilon}{2})$ equals

$$T_{CE}(\alpha_1, \alpha_2, \epsilon)$$
$$= -\frac{\alpha_1}{\frac{\alpha_1 + \alpha_2}{2} - \frac{\epsilon}{2} + \epsilon} + \frac{1 - \alpha_1}{1 - \frac{\alpha_1 + \alpha_2}{2} + \frac{\epsilon}{2} - \epsilon} - \frac{\alpha_2}{\frac{\alpha_1 + \alpha_2}{2} - \frac{\epsilon}{2}} + \frac{1 - \alpha_2}{1 - \frac{\alpha_1 + \alpha_2}{2} + \frac{\epsilon}{2}}$$
$$= 2\Big(\frac{1 - \alpha_1}{1 - \alpha_1 - \alpha_2 - \epsilon} - \frac{\alpha_1}{\alpha_1 + \alpha_2 + \epsilon} + \frac{1 - \alpha_2}{1 - \alpha_1 - \alpha_2 + \epsilon} - \frac{\alpha_2}{\alpha_1 + \alpha_2 - \epsilon}\Big). \tag{43}$$

Given that $0.5 <= \alpha_2 < \alpha_1 < 1$ and $0 \leq \epsilon < \alpha_1 - \alpha_2$ (by $\delta < \|q_1 - q_2\|_2^2$), we can determine the sign of $T_{CE}(\alpha_1, \alpha_2, \epsilon)$.

**Lemma A.1.** $T_{CE}(\alpha_1, \alpha_2, \epsilon) < 0$. *Therefore, when $\delta < \|\boldsymbol{q}_1 - \boldsymbol{q}_2\|_2^2$, the CE loss always yields negative $\beta$ scores.*

*Proof.* We denote that

$$
\begin{cases}
D_1 = 2 - \alpha_1 - \alpha_2 - \epsilon > 0, \\
D_2 = \alpha_1 + \alpha_2 + \epsilon > 0, \\
D_3 = 2 - \alpha_1 - \alpha_2 + \epsilon > 0, \\
D_4 = \alpha_1 + \alpha_2 - \epsilon > 0,
\end{cases}
\tag{44}
$$

Then, we have

$$
\begin{aligned}
\frac{1}{2} T_{CE}(\alpha_1, \alpha_2, \epsilon) &= \frac{1 - \alpha_1}{D_1} - \frac{\alpha_1}{D_2} + \frac{1 - \alpha_2}{D_3} - \frac{\alpha_2}{D_4} \\
&= \frac{(1 - \alpha_1)D_2 - \alpha_1 D_1}{D_1 D_2} + \frac{(1 - \alpha_2)D_4 - \alpha_2 D_3}{D_3 D_4} \\
&= \frac{-\alpha_1 + \alpha_2 + \epsilon}{D_1 D_2} + \frac{\alpha_1 - \alpha_2 - \epsilon}{D_3 D_4} \\
&= (\alpha_1 - \alpha_2 - \epsilon)\left(\frac{1}{D_3 D_4} - \frac{1}{D_1 D_2}\right) \\
&= (\alpha_1 - \alpha_2 - \epsilon)\left(\frac{D_1 D_2 - D_3 D_4}{D_1 D_2 D_3 D_4}\right)
\end{aligned}
\tag{45}
$$

where $\alpha_1 - \alpha_2 - \epsilon > 0$, $D_1 D_2 D_3 D_4 > 0$, and $D_1 D_2 - D_3 D_4 = -4c(a + b - 1) < 0$. Therefore, $h(\frac{\alpha_1 + \alpha_2}{2} - \frac{\epsilon}{2}) = T_{CE}(\alpha_1, \alpha_2, \epsilon) < 0$. Based on the previous deductions, we can conclude that $\beta$ is always negative when $\delta < \|\boldsymbol{q}_1 - \boldsymbol{q}_2\|_2^2$.

**Focal Loss.** Due to the complexity of Focal loss, we specifically analyze the case when $\gamma_{FL} = 1$. The commonly used $\gamma$s are larger and the balance scores become even greater empirically for these values. It's worth noting that $\delta \geq \|\boldsymbol{q}_1 - \boldsymbol{q}_2\|_2^2$ doesn't guarantee the Focal loss to yield $\boldsymbol{p}_i^* = \boldsymbol{q}_i$ since the Focal loss is not strictly proper (Charoenphakdee et al., 2021). Nevertheless, its behavior with $\delta < \|\boldsymbol{q}_1 - \boldsymbol{q}_2\|_2^2$ can be sufficiently clear. We mainly focus on these cases in the following proofs.

Similar to CE loss, the sign of $\beta$ is associated with function

$$
\begin{aligned}
g(x) &= (1 - \alpha_1)\left(\frac{x + \epsilon}{1 - x - \epsilon} - \log(1 - x - \epsilon)\right) - \alpha_1\left(\frac{1 - x - \epsilon}{x + \epsilon} - \log(x + \epsilon)\right) \\
&+ (1 - \alpha_2)\left(\frac{x}{1 - x} - \log(1 - x)\right) - \alpha_2\left(\frac{1 - x}{x} - \log(x)\right).
\end{aligned}
\tag{46}
$$

Note that $g(x)$ also increases monotonously. Therefore, the sign of $\beta$ at the optimal $\boldsymbol{p}_i^*$ is the same as the sign of $g(\frac{\alpha_1 + \alpha_2}{2} - \frac{\epsilon}{2})$, which equals

$$
\begin{aligned}
T_{FL}(\alpha_1, \alpha_2, \epsilon) &= (1 - \alpha_1)\left(\frac{E}{1 - E} - \log(1 - E)\right) - \alpha_1\left(\frac{1 - E}{E} - \log(E)\right) \\
&+ (1 - \alpha_2)\left(\frac{F}{1 - F} - \log(1 - F)\right) - \alpha_2\left(\frac{1 - F}{F} - \log(F)\right),
\end{aligned}
\tag{47}
$$

where $E = \frac{\alpha_1 + \alpha_2 + \epsilon}{2}$, $F = \frac{\alpha_1 + \alpha_2 - \epsilon}{2}$, constrained by $\alpha_2 < F \leq E < \alpha_1$, and $E + F = \alpha_1 + \alpha_2$.

**Lemma A.2.** $T_{FL}(\alpha_1, \alpha_2, \epsilon) > 0$. *Therefore, when $\delta < \|\boldsymbol{q}_1 - \boldsymbol{q}_2\|_2^2$, the FL loss with $\gamma_{FL} = 1$ always yields positive $\beta$ scores.*

*Proof.* We rewrite $T_{FL}(\alpha_1, \alpha_2, \epsilon)$ as

$$
\begin{aligned}
T_{FL}(\alpha_1, \alpha_2, \epsilon) &= \mu(\alpha_2, x) + \mu(\alpha_1, -x), \\
\mu(w, x) &= (1 - w)\left(\frac{w + x}{1 - w - x} - \log(1 - w - x)\right) \\
&- w\left(\frac{1 - w - x}{w + x} - \log(w + x)\right),
\end{aligned}
\tag{48}
$$

where $w \in [0.5, 1.0)$ and $x \in (0, \min(\frac{\alpha_1 - \alpha_2}{2}, 1 - w)]$. We show 2 crucial properties of $\mu(w, x)$:

a) $\mu(w, -x) + \mu(0.5, x) > 0$.

*Proof.* We reformulate the left part of the inequality as

$$
\begin{aligned}
&\mu(w, -x) + \mu(0.5, x) \\
=&(1 - w)\left(\frac{w - x}{1 - w + x} - \log(1 - w + x)\right) - w\left(\frac{1 - w + x}{w - x} - \log(w - x)\right) \\
&+ 0.5\left(\frac{0.5 + x}{0.5 - x} - \log(0.5 - x)\right) - 0.5\left(\frac{0.5 - x}{0.5 + x} - \log(0.5 + x)\right) \\
=& -(1 - w)\log(1 - w + x) + w\log(w - x) - 0.5\log(0.5 - x) + 0.5\log(0.5 + x) \\
&+ (1 - w)\frac{w - x}{1 - w + x} - w\frac{1 - w + x}{w - x} + 0.5\frac{0.5 + x}{0.5 - x} - 0.5\frac{0.5 - x}{0.5 + x},
\end{aligned}
\tag{49}
$$

where we consider

$$T_1 = w\log(w - x) - 0.5\log(0.5 - x), \tag{50}$$
$$T_2 = 0.5\log(0.5 + x) - (1 - w)\log(1 - w + x). \tag{51}$$

to find that

$$\frac{\partial T_1}{\partial x} = \frac{0.5}{0.5 - x} - \frac{w}{w - x} = \frac{(w - 0.5)x}{(0.5 - x)(w - x)} \geq 0, \tag{52}$$

$$\frac{\partial T_2}{\partial x} = \frac{0.5}{0.5 + x} - \frac{1 - w}{1 - w + x} = \frac{(w - 0.5)x}{(0.5 + x)(1 - w + x)} \geq 0. \tag{53}$$

$$T_1 + T_2 > (T_1 + T_2)\big|_{x=0} = \phi(w) = w\log w - (1 - w)\log(1 - w). \tag{54}$$

Because $\phi''(w) = \frac{1 - 2w}{w(1 - w)} \leq 0$, $T_1 + T_2 > \phi(w) \geq \min(\phi(0.5), \phi(1^-)) = 0$. Now that we have considered the sum of logarithmic terms in Eq. (49), we proceed to inspect the sum of the rest terms:

$$
\begin{aligned}
&(1 - w)\frac{w - x}{1 - w + x} - w\frac{1 - w + x}{w - x} + 0.5\frac{0.5 + x}{0.5 - x} - 0.5\frac{0.5 - x}{0.5 + x} \\
>&(1 - w)\frac{0.5 - x}{0.5 + x} - w\frac{1 - w + x}{w - x} + 0.5\frac{0.5 + x}{0.5 - x} - 0.5\frac{0.5 - x}{0.5 + x} \\
=& F(w, x)
\end{aligned}
\tag{55}
$$

where we can find

$$
\begin{aligned}
\frac{\partial F(w, x)}{\partial w} &= -\frac{0.5 - x}{0.5 + x} - \frac{1 - w + x}{w - x} - w\frac{x - w - (1 - w + x)}{(w - x)^2} \\
&= -\frac{0.5 - x}{0.5 + x} - \frac{1 - w + x}{w - x} + \frac{w}{(w - x)^2} \\
&= \frac{w - (w - x)(1 - w + x)}{(w - x)^2} - \frac{0.5 - x}{0.5 + x} \\
&= \frac{x + (w - x)^2}{(w - x)^2} - \frac{0.5 - x}{0.5 + x} \\
&= 1 + \frac{x}{(w - x)^2} - \left(1 - \frac{2x}{0.5 + x}\right) \\
&= \frac{x}{(w - x)^2} + \frac{2x}{0.5 + x} > 0
\end{aligned}
\tag{56}
$$

$$F(w, x) > F(0.5, x) = 0. \tag{57}$$

Therefore, Eq. (49) is positive because the sum of all its terms is positive.

b) $\mu(w, x) > \mu(0.5, x)$

*Proof.*

$$
\begin{aligned}
&\mu(w, x) - \mu(0.5, x) \\
&= -(1-w)\log(1-w-x) + w\log(w+x) + 0.5\log(0.5-x) - 0.5\log(0.5+x) \\
&\quad + (1-w)\frac{w+x}{1-w-x} - w\frac{1-w-x}{w+x} - 0.5\frac{0.5+x}{0.5-x} + 0.5\frac{0.5-x}{0.5+x},
\end{aligned}
\tag{58}
$$

where we consider

$$
\begin{aligned}
T_3 = &-(1-w)\log(1-w-x) + w\log(w+x) \\
&+ 0.5\log(0.5-x) - 0.5\log(0.5+x),
\end{aligned}
\tag{59}
$$

$$
T_4 = (1-w)\frac{w+x}{1-w-x} - w\frac{1-w-x}{w+x} - 0.5\frac{0.5+x}{0.5-x} + 0.5\frac{0.5-x}{0.5+x}.
\tag{60}
$$

We can find that

$$
\begin{aligned}
\frac{\partial T_3}{\partial x} =& \frac{1-w}{1-w-x} + \frac{w}{w+x} - \frac{0.5}{0.5-x} - \frac{0.5}{0.5+x} \\
=& \frac{0.5(1-w) - x(1-w) - 0.5(1-w) + 0.5x}{(1-w-x)(w+x)} + \frac{0.5w + wx - 0.5w - 0.5x}{(w+x)(0.5+x)} \\
=& \frac{(w-0.5)x}{(1-w-x)(w+x)} + \frac{(w-0.5)x}{(w+x)(0.5+x)} \\
>& 0
\end{aligned}
\tag{61}
$$

$$
T_3 > T_3\big|_{x=0} = w\log w - (1-w)\log(1-w) > 0.
\tag{62}
$$

Meanwhile, we can also find

$$
\begin{aligned}
T_4 =& (1-w)\frac{w+x}{1-w-x} - w\frac{1-w-x}{w+x} - 0.5\frac{0.5+x}{0.5-x} + 0.5\frac{0.5-x}{0.5+x} \\
>& (1-w)\frac{w+x}{1-w-x} - w\frac{0.5-x}{0.5+x} - 0.5\frac{0.5+x}{0.5-x} + 0.5\frac{0.5-x}{0.5+x} \\
=& G(w, x).
\end{aligned}
\tag{63}
$$

where we have

$$
\begin{aligned}
\frac{\partial G(w, x)}{\partial w} =& -\frac{w+x}{1-w-x} + (1-w)\frac{1-w-x+w+x}{(1-w-x)^2} - \frac{0.5-x}{0.5+x} \\
=& \frac{1-w-(w+x)(1-w-x)}{(1-w-x)^2} - \frac{0.5-x}{0.5+x} \\
=& \frac{(1-w-x)^2 + x}{(1-w-x)^2} - \frac{0.5-x}{0.5+x} \\
=& \frac{x}{(1-w-x)^2} + \frac{2x}{0.5+x} > 0,
\end{aligned}
\tag{64}
$$

which results in $T_4 > G(w, x) > G(0.5, x) = 0$.

Therefore, combining a) and b), we have $-\mu(\alpha_1, -x) < \mu(0.5, x) < \mu(\alpha_2, x)$, *i.e.*, $T_{FL}(\alpha_1, \alpha_2, \epsilon) = \mu(\alpha_2, x) + \mu(\alpha_1, -x) > 0$.

## B. More Details for Experiments

### B.1. Compared Baselines

We adopt diverse training-time methods for comparison including the vanilla CE loss. We set hyperparameters for the compared methods following (Noh et al., 2023). Specifically, we compare calibration performance with the traditional

regularization-based methods including a) ECP (Pereyra et al., 2017) with 0.1 as the coefficient for entropy penalty, b) MMCE (Kumar et al., 2018), c) LS (Müller et al., 2019) with $\alpha = 0.05$, d) mbLS (Liu et al., 2022) with $m = 6$ for CIFAR10/100 and $m = 10$ for Tiny-ImageNet, e) FL (Ross & Dollár, 2017) with fixed $\gamma = 3$, f) FLSD (Mukhoti et al., 2020) adopting $\gamma$ schedule of FLSD-53 variant, g) CPC (Cheng & Vasconcelos, 2022), and h) FCL (Liang et al., 2024) with $\gamma = 3$ and $\lambda = 0.5$. We also compare our CSM with data-driven calibration methods, *i.e.*, a) Mixup (Zhang et al., 2018) with a Beta distribution shape parameter $\alpha = 0.2$, b) RegMixup (Pinto et al., 2022) with a Beta distribution shape parameter $\alpha = 10$, c) AugMix (Hendrycks et al., 2019) with a coefficient of 1.0 for JS Divergence term, which yields better accuracy compared to original value of 12, and d) RankMixup (M-NDCG) (Noh et al., 2023) with weight for M-NDCG set as 0.1 and shape parameters $\alpha = 1, 2$ for CIFAR10/100 and TinyImageNet, respectively.

### B.2. Datasets and Augmented Samples

We adopt CIFAR-10, CIFAR-100, and Tiny-ImageNet as the evaluated datasets. We hold the principle that the ratios of generated sample amount versus training size are fixed to be less than a certain value across different datasets, *e.g.*, 4.0 in our experiments. Note that the number of augmented samples we use in training is much less than that in Mixup methods. Meanwhile, we fix the size of each sample set generated with the same noise $x^T$ as 8 for all datasets. Here are detailed descriptions of the datasets and augmented samples:

**CIFAR-10**: The CIFAR-10 dataset contains $60,000$ RGB images of the size $32 \times 32$. All images fall into one of 10 semantic categories. By default, the dataset is split as $50,000$, $5,000$, and $5,000$ samples for training, validation, and testing. We generate 550 sets for each of the 45 class pairs, producing $198,000$ samples for confidence-aware augmentation.

**CIFAR-100**: The CIFAR-100 dataset consists of $60,000$ $32 \times 32$ color images in 100 classes. The split is similar to CIFAR-10 with $50,000$ for training, $5,000$ for validation, and $5,000$ for testing. We generate 5 sample sets for each distinctive class pair, yielding $198,400$ augmented samples in total.

**Tiny-ImageNet**: The Tiny-ImageNet dataset is a subset of the large-scale ImageNet dataset and includes $120,000$ images of a large set of 200 classes. Each image is a downsized ImageNet sample of size $64 \times 64$. For CSM augmentation, we generate 2 sets for every $19,900$ class pair, producing a total of $318,400$ samples.

The generation process takes as much or even less time than the training time. Specifically, it takes less than 4h for a single A4000 GPU to generate the amount of all CIFAR-10 or CIFAR-100 augmented samples, while using less than 8h for the same computing units to generate for Tiny-ImageNet.

### B.3. Evaluation Metrics

To assess the model calibration, we employ four key metrics: Expected Calibration Error (ECE), Adaptive Expected Calibration Error (AECE), Overconfidence Error (OE), and Underconfidence Error (UE).

ECE approximates the average discrepancy between a model's confidence and accuracy. Practically, it is estimated across equally spaced confidence bins. Let $B_1, \ldots, B_M$ denote $M$ bins partitioning predictions into intervals $[0, \frac{1}{M}), \ldots, [\frac{M-1}{M}, 1]$. For each bin $B_m$, the accuracy and confidence are aomputed as $\text{acc}(B_m) = \frac{1}{|B_m|} \sum_{i \in B_m} \mathbb{I}(y_i = \hat{y}_i)$ and $\text{conf}(B_m) = \frac{1}{|B_m|} \sum_{i \in B_m} \hat{p_i}$, respectively, where $y_i$ is the true label, $\hat{y}_i$ is the predicted label, and $\hat{p_i}$ is the predicted probability for the winning class. ECE is then defined as

$$\text{ECE} = \sum_{m=1}^{M} \frac{|B_m|}{N} \left| \text{acc}(B_m) - \text{conf}(B_m) \right|, \tag{65}$$

where $N$ is the total number of samples. Lower ECE values indicate better calibration.

AECE addresses potential biases from fixed-width binning by constructing bins with adaptive widths to ensure equal *sample counts* per bin. This approach reduces sensitivity to irregular confidence distributions. The calculation mirrors ECE by using Eq. (65) but uses bins $B_1, \ldots, B_M$ where each bin contains approximately $N/M$ samples.

OE isolates cases where the model's confidence exceeds its accuracy compared to ECE. Using the same binning as ECE, OE

*Table 8.* Evaluation results with Wide-ResNet-26-10 on CIFAR-10 and Tiny-ImageNet.

| Method | CIFAR-10 | | | Tiny-ImageNet | | |
| | ACC↑ | ECE↓ | AECE↓ | ACC↑ | ECE↓ | AECE↓ |
| --- | --- | --- | --- | --- | --- | --- |
| CE | 95.80 | 2.70 | 2.66 | 65.18 | 6.08 | 6.06 |
| Mixup | **96.53** | 3.14 | 3.08 | 66.36 | 3.77 | 3.75 |
| MbLS | 95.70 | 1.45 | 2.78 | 65.30 | 2.57 | 2.32 |
| RegMixup | 95.44 | 4.18 | 3.99 | 63.40 | 3.87 | 3.93 |
| FCL | 95.84 | 0.92 | 1.39 | 64.62 | 6.85 | 6.85 |
| RankMixup | 95.73 | 1.62 | 1.53 | 65.56 | 3.83 | 3.94 |
| CSM | 96.09 | **0.49** | **0.23** | **67.81** | **1.67** | **1.66** |

*Table 9.* Evaluation results with DenseNet-121 on CIFAR-10.

| Metric | CE | MbLS | Mixup | RegMixup | FCL | RankMixup | CSM |
| --- | --- | --- | --- | --- | --- | --- | --- |
| ACC↑ | 94.68 | 94.91 | 95.28 | **96.23** | 95.41 | 94.66 | 95.71 |
| ECE↓ | 3.37 | 1.64 | 2.01 | 5.60 | 0.66 | 2.87 | **0.51** |
| AECE↓ | 3.31 | 3.52 | 2.15 | 5.39 | 1.28 | 2.84 | **0.19** |

*Table 10.* Results on long-tailed datasets.

| Method | CIFAR10-LT | | | | CIFAR100-LT | | | |
| | $\rho = 10$ | | $\rho = 100$ | | $\rho = 10$ | | $\rho = 100$ | |
| | ACC↑ | ECE↓ | ACC↑ | ECE↓ | ACC↑ | ECE↓ | ACC↑ | ECE↓ |
| --- | --- | --- | --- | --- | --- | --- | --- | --- |
| CE | 86.39 | 6.60 | 70.36 | 20.53 | 55.70 | 22.85 | 38.32 | 38.23 |
| Mixup | 87.10 | 6.55 | 73.06 | 19.20 | 58.02 | 19.69 | 39.54 | 32.72 |
| Remix | 88.15 | 6.81 | 75.36 | 15.38 | 59.36 | 20.17 | 41.94 | 33.56 |
| UniMix | 89.66 | 6.00 | 82.75 | 12.87 | 61.25 | 19.38 | 45.45 | 27.12 |
| RankMixup | 89.80 | 5.94 | 75.41 | 14.10 | **63.83** | 9.99 | 43.00 | 18.74 |
| CSM | **90.97** | **1.81** | **86.22** | **3.73** | 62.35 | **7.45** | **48.91** | **16.02** |

is defined as

$$\mathrm{OE} = \sum_{m=1}^{M} \frac{|B_m|}{N} \max\left(0, \mathrm{conf}(B_m) - \mathrm{acc}(B_m)\right). \tag{66}$$

This metric evaluates overconfident predictions, with lower values indicating fewer instances of excessive confidence.

Conversely, UE captures scenarios where the model's confidence underestimates its accuracy. It is computed as

$$\mathrm{UE} = \sum_{m=1}^{M} \frac{|B_m|}{N} \max\left(0, \mathrm{acc}(B_m) - \mathrm{conf}(B_m)\right). \tag{67}$$

Lower UE values suggest better alignment in underconfident cases.

## C. More Experimental Results

**Network Architecture** We provide results on Wide-ResNet-26-10 and DenseNet-121 to verify the performance consistency of our method in Table 8 and Table 9.

**Long-tailed Datasets** We also analyze the effectiveness of CSM on long-tailed datasets, where the challenges of class imbalance and miscalibration are more severe. Following the commonly-adopted setup in (Xu et al., 2021; Zhong et al., 2021), ResNet32 network and CIFAR10/100-LT datasets are adopted as the backbone and benchmarks, respectively. As displayed in Table 10, our method outperforms the vanilla mixup and existing mixup-based LT approaches (Chou et al., 2020; Xu et al., 2021; Noh et al., 2023) across datasets and imbalance factors ($\rho = 10$ and $\rho = 100$) in terms of both ACC and ECE, demonstrating competitive or superior effectiveness without particular designs to learn on long-tailed datasets. These results suggest that our model's authentic augmentations and confidence-aware data are key factors contributing to its superior performance, even in highly imbalanced LT settings. Our model's significant improvements in both accuracy and calibration metrics demonstrate its robustness in handling long-tailed datasets.

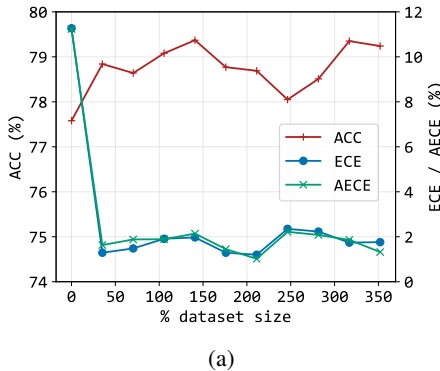

(a)

*Figure 7.* **Analysis of the amount of augmented data**: Classification ACC, ECE, and AECE on CIFAR-100 *w.r.t* different number of augmented samples relative to the dataset size.

*Table 11.* **Calibration Comparison with Different Number of Augmented Samples** *per Training Sample.*

| $N_{\text{AUGS}}$ | 1 | 2 | 3 |
|---|---|---|---|
| ECE: CIFAR-10 | 0.83 | 0.54 | 0.39 |
| ECE: CIFAR-100 | 2.07 | 1.29 | 1.74 |

*Table 12.* **Comparison of the Estimated Training Time.**

| METHOD | CE | MBLS | MIXUP | REGMIXUP | RANKMIXUP | OURS | OURS(EQ-DATA) |
|---|---|---|---|---|---|---|---|
| TRAINING TIME | 2.63H | 2.65H | 3.48H | 3.50H | 4.30H | 4.28H | 2.64H |

**Number of Augmented Samples** We plot the calibration results *w.r.t.* different amount of used augmentations in Figure 7(c). To ensure equal comparison, we randomly sample dataset samples and augmented ones with the fixed ratio 1:2 during training regardless of the available amount of augmentations. Surprisingly, the model can calibrate well even with relatively small proportions of augmentations. Meanwhile, ECE and AECE values are jointly minimized with augmentation amount around 200% dataset size, reaching their best value of 1.20 and 1.02 on CIFAR-100. These results effectively validate the consistency and reliability of our method.

**Number of Augmented Samples per Training Sample** We provide more analyses about the number of augmented samples *per training sample* (denoted as $N_{aug}$) with results in Table 11. It can be observed that adding the number of accompanied augmentations per dataset sample can generally improve the final calibration performance. This is because a larger number of $N_{augs}$ can sample more sufficient proximal data for training, better filling the domain space and providing more accurate confidence estimation. However, simply using larger $N_{augs}$s could also raise the computational overhead and slow down the training.

**Estimated Training Time** We provide a comparison of the estimated training time in Table 12. One can see the number of augmented samples per batch is the main factor influencing the training time. CSM outperforms others in calibration while maintaining reasonable speed. Even with equalized training samples, *i.e.*, EQ-DATA, it achieves competitive calibration performance. Augmented samples need no re-generation across model, objective, or annotation changes, enabling efficient modular study. We run CSM with a single RTX A4000 device.

