# OpenReview forum: "Beyond One-Hot Labels: Semantic Mixing for Model Calibration"
_ICML.cc/2025/Conference — ICML 2025 poster_

### Official Review · Reviewer_jJxJ · 2025-03-09

**Overall Recommendation:** 4

**Summary:**

This paper propose Calibration-aware Semantic Mixing, a model calibration approach using diffusion-based data augmentation, like “semantic mixup”.Unlike traditional one-hot labeling, CSM generates mixed samples with soft labels with the CLIP. The authors introduce a reannotation technique using CLIP features and investigate the influence of loss functions prove L2 loss is good for enhances calibration.

**Claims And Evidence:**

1. CSM improves model calibration by introducing semantically meaningful augmentations, validated through ECE and AECE reductions.
2.  L2 loss leads to balanced learning, demonstrated by both theoretical insights and empirical results.

**Essential References Not Discussed:**

N/A

**Experimental Designs Or Analyses:**

1. extensive comparisons with other  train time calibration techniques.
2. AUROC (%) for robustness evaluation under distribution shifts.
3. Calibration performance with post hoc calibration methods.
4. Reliability diagrams and ablation studies

**Methods And Evaluation Criteria:**

1. The paper introduces a diffusion-based augmentation method using L2 loss supported by theoretical evidence.
2. This paper is evaluate with ECE, AECE on CIFAR-10, CIFAR-100, and Tiny-ImageNet, using ResNet-50/101, Wide-ResNet-26-10 and DenseNet-12

**Other Comments Or Suggestions:**

1. Provide comparisons of computational cost with existing calibration techniques.
2. Include failure cases where CSM does not improve calibration performance

**Other Strengths And Weaknesses:**

Weaknesses:
1. Limited discussion on computational efficiency
2. Hyperparameter sensitivity analysis is not well explored.
3. No Transformer model is included for comparison

**Questions For Authors:**

1. How does the choice of diffusion model affect the performance of CSM? How about we use other generative models. There are some  generative model doing interpolation , like GANs. Does it helps?
2. How does CSM compare to Mixup in terms of training efficiency and memory usage?

**Relation To Broader Scientific Literature:**

This paper is highly related to the field of calibration, with the relationship to test time augmentation and label smoothing.

**Theoretical Claims:**

The authors provide a theoretical justification for their reannotation strategy and choice of loss functions

---

> ### Author Rebuttal · Authors · 2025-03-31
>
> ## Response to Reviewer jJxJ
>
> Thanks for your helpful suggestions! Here’s our response:
>
> **Q4-1**: Training computational cost and memory usage compared to existing methods.
>
> **A4-1**: As also analyzed in **A2-3**, we compare the computational efficiency in terms of the training time in **A2-3 Table B**:
>
> We can conclude that the **number of augmented samples per batch** is the **major factor** in training time. Our CSM maintain relative efficiency considering the calibration effectiveness. When compared on the EQ-DATA setting in the main paper Table 2, CSM achieves competitive calibration results with an equalized training time.
>
> Such factor also decides memory usage. CSM uses ~$\frac{3}{2}$ memory compared to Mixup/RegMixup, while using ~equal memory compared to the RankMixup (depending on setep).
>
> Regarding resource consumption, we generate augmented samples and train our model on one A4000 device. It's worth noting that CSM **needs no re-generation** when switching model/objectives/re-annotation methods, making it more efficient for decoupled study of these modules.
>
> We will include the key information in the main paper.
>
> **Q4-2**: Hyperparameter sensitivity.
>
> **A4-2**:
> We have analyzed hyperparameter $s$ and the number of augmented samples in Appendix C. We provide more analyses about **the number of augmented samples per training sample** (denoted as $N_{aug}$) here.
> **Table C:**
> | $N_{augs}$ | 1 | 2 | 3 |
> | - | - | - | - |
> | ECE: CIFAR-10 | 0.83 | 0.54 | 0.39 |
> | ECE: CIFAR-100 | 2.07 | 1.29 | 1.74 |
>
> It can be observed that adding the number of accompanied augmentations per dataset sample can generally improve the final calibration performance. This is because a larger number of $N_{augs}$ can sample more sufficient proximal data for training, better filling the domain space and providing more accurate confidence estimation.
>
> For **sensitivity analysis**, we have the following key observations to enhance our experiments considering **existing results** in Appendix C:
> 1. Our proposed CSM is not sensitive to the total number of augmented samples. A relatively smaller quantity can still make CSM effective.
> 2. CSM is relatively sensitive to the scaling factor $s$ as it is related to the temperature. Nevertheless, under an appropriate range of $s$, the method can perform consistently well.
>
> **Q4-3**: Evaluation on the transformer architecture.
>
> **A4-3**: Evaluations on the Swin-Transformer architecture verify our method's equal or stronger effectiveness. Please refer to **A1-3 Table A**.
>
> **Q4-4**: Failure cases of CSM.
>
> **A4-4**: While our method can perform well on all the evaluated datasets, we admit there are some cases CSM fails. For instance, although CSM surpass compared methods on ECE/AECE, some post-temperature results on the CIFAR-100 dataset are not comparative to the SOTA methods. Specifically, they exhibit balanced pre- and post-temperature results (searched $T = 1$) with remarkable pre-temperature ECE values but slightly larger calibration errors after temperature scaling. Such phenomenon is also stated in the calibration literature [1], where they found a balance exists in the two results. Considering both results together, our CSM can still achieve satisfactory  model calibration.
>
> **Q4-5**: Will the choice of generative models affect the performance?
>
> **A4-5**:
> The choice of the generative model does influence the final prediction results. Some existing study [2] has already proved that adopting a better generative model could improve the classification model's robustness. We anticipate that a better generative backbone can achieve superior confidence calibration using our CSM.
> This aligns with our empirical experiences that using the ordinary Stable Diffusion architecture, we can only achieve suboptimal results as shown in the table:
> | Gen. Model | ACC | ECE$\downarrow$ | AECE$\downarrow$ |
> | - | - | - | - |
> |SD [3]|76.87|1.93|1.78|
> |EDM [4] (in our CSM)|**78.84**|**1.29**|**1.63**|
>
> Therefore, we anticipate that a typical GAN with less parameters or lower fidelity compared to diffusion models would yield worse results.
>
> **References**
>
> [1] Wang, D. B., Feng, L., & Zhang, M. L. (2021). Rethinking calibration of deep neural networks: Do not be afraid of overconfidence. Advances in Neural Information Processing Systems, 34, 11809-11820.
>
> [2] Wang, Z., Pang, T., Du, C., Lin, M., Liu, W., & Yan, S. (2023, July). Better diffusion models further improve adversarial training. In International conference on machine learning (pp. 36246-36263). PMLR.
>
> [3] Rombach, R., Blattmann, A., Lorenz, D., Esser, P., & Ommer, B. (2022). High-resolution image synthesis with latent diffusion models. In Proceedings of the IEEE/CVF conference on computer vision and pattern recognition (pp. 10684-10695).
>
> [4] Karras, T., Aittala, M., Aila, T., & Laine, S. (2022). Elucidating the design space of diffusion-based generative models. Advances in neural information processing systems, 35, 26565-26577.

---

### Official Review · Reviewer_WPRF · 2025-03-12

**Overall Recommendation:** 3

**Summary:**

This paper introduces a novel framework, Calibration-aware Semantic Mixing (CSM), designed to improve model calibration. The key contribution lies in addressing the limitations of one-hot labeled datasets by proposing a data augmentation technique that leverages semantic mixing to generate diverse samples via diffusion models. The paper also introduces reannotation techniques to enhance confidence annotation accuracy and explores different loss functions to achieve confidence-balanced learning. Experimental results demonstrate that CSM surpasses existing calibration methods, delivering superior performance across multiple benchmarks and tasks.

## update after rebuttal
My concerns have been addressed and would like to recommend accept.

**Claims And Evidence:**

The claims presented in the paper are well-supported by empirical evidence from the experiments.

**Essential References Not Discussed:**

None.

**Experimental Designs Or Analyses:**

The overall experimental design and analysis are well-structured and reasonable. The authors compare the proposed calibration technique against several widely-used calibration algorithms on diverse models and datasets. Ablation studies effectively highlight the contributions of different components.
However, I have two major concerns:
- Calibrated Reannotation – The authors utilize CLIP’s visual encoder for reannotation, but they do not discuss or compare it with a simple baseline that directly adopts CLIP outputs as labels. Evaluating this baseline would help assess the added benefit of the proposed reannotation approach.
- Calibration-aware Data Augmentation – The study proposes a semantic mixing strategy for generating calibrated samples using diffusion models. However, a crucial baseline is missing: directly using generated images from diffusion models without semantic mixing. Evaluating this approach would provide a clearer understanding of semantic mixing’s contribution.

**Methods And Evaluation Criteria:**

The proposed method is both novel and well-motivated, offering fresh insights into model calibration. The evaluation follows standard practice in this field, employing accuracy, Expected Calibration Error (ECE), and post-temperature scaling as key metrics.

**Other Comments Or Suggestions:**

None.

**Other Strengths And Weaknesses:**

Overall, the paper is well-written and easy to follow. The proposed semantic mixing framework is conceptually sound and coherently presented. The evaluation is comprehensive, and the method demonstrates superior calibration performance compared to existing baselines. However, as noted earlier, some concerns remain regarding the theoretical claims (Proposition 3.4) and the experimental design (missing baselines for reannotation and augmentation methods).

**Questions For Authors:**

- Could you discuss or evaluate simple baselines for calibrated reannotation, such as directly adopting CLIP outputs as labels?
- Could you discuss or evaluate simple baselines for calibration-aware data augmentation, such as using generated images without semantic mixing?
- Could you clarify Proposition 3.4, particularly regarding why L2 loss outperforms cross-entropy and focal loss?

**Relation To Broader Scientific Literature:**

Addressing model calibration from a data-driven perspective is an interesting and promising direction. The results across different models and datasets suggest strong potential for real-world applications. The proposed algorithm could contribute significantly to the field of trustworthy AI, enhancing model reliability and confidence estimation in diverse applications.

**Theoretical Claims:**

One concern arises in Proposition 3.4, where the paper claims that L2 loss outperforms cross-entropy (CE) and focal loss. The reasoning behind this claim is unclear and requires further clarification.

---

> ### Author Rebuttal · Authors · 2025-03-31
>
> ## Response to Reviewer WPRF
>
> Thank you for your encouraging feedback on the clarity, soundness, and comprehensive evaluation of our work. We truly appreciate your thoughtful suggestions for clarity and comprehensive validation. Here are our responses to the suggestions:
>
> **Q3-1**: Clarify Proposition 3.4 (L2 loss vs. CE/FL).
>
> **A3-1**: Thank you for commenting on the clarity issue. We need to clarify that there exists a **typo** in Proposition 3.4 potentially hindered understanding. Proposition 3.4 should have been presented as
>
> - $\forall \delta \ge 0$, we have $\beta(p^{L2}_1, p^{L2}_2) = 0$,
>
> which is an equation rather than an inequality for the $\mathcal{L}_2$ loss's balance function, meaning that when two similar samples exceed the model's discriminability, $\mathcal{L}_2$ loss tends to **balance the learned labels** of the harder and softer instances, instead of tending to fit a specific one of them. Note that the proof of Proposition 3.4 we provided in Appendix A.3 does prove that $\beta(p^{L2}_1,p^{L2}_2) = 0$.
>
> As proved in Appendix A, easier samples with $\delta \ge \|q^{L2}_1, q^{L2}_2\|$ would generally have $\|p^{L2}_i, q^{L2}_i\| = 0, i=1,2$. Smaller $\delta$s indicate difficulty for the learned model to separate the outputs for both samples, hence introducing a balancing problem. A theoretically non-zero balancing score $\beta$ means one of $\|p^{L2}_i, q^{L2}_i\|, i=1,2$, is minimized more completely, indicating the imbalanced nature of the objective.
>
> Among the three, only the L2 loss theoretically zeros the $\beta$ balancing score, indicating its ability to balance over- and under-confidence of difficult soft-labeled sample pairs, hence being a superior loss for calibration with our augmented samples. In practice, the soft-label distribution may disturb such balance, which can be one of our future study.
>
> We will correct this typo in the revised main paper.
>
> **Q3-2**: Baseline for directly adopting CLIP labels. (Var. 1)
>
> **Q3-3**: Baseline for diffusive sample augment without semantic mixing. (Var. 2)
>
> **A3-2, 3-3**:
> We evaluate these two variants and compare them with our proposed CSM as follows:
> CIFAR-10:
> | Variant | ACC | ECE$\downarrow$ | AECE$\downarrow$ |
> | - | - | - | - |
> |Var. 1|*92.13*|2.68(0.92)|2.67(0.88)|
> |Var. 2|**96.12**|2.45(0.96)|2.44(1.13)|
> |CSM (Ours)|95.79|**0.54(0.54)**|**0.33(0.33)**|
>
> CIFAR-100:
> | Variant | ACC | ECE$\downarrow$ | AECE$\downarrow$ |
> | - | - | - | - |
> |Var. 1|*66.60*|52.78(1.36)|52.78(**1.11**)|
> |Var. 2|**79.24**|10.84(2.48)|10.84(2.41)|
> |CSM (Ours)|78.84|**1.29(1.29)**|**1.63**(1.63)|
>
> From these results, we can have these valuable observations:
> 1. The vanilla CLIP annotation method yields the worst ACC and pre-temperature calibration errors, primarily due to the noisy information by annotating all classes. Such degradation is significant for CIFAR-100, in which there are more classes so that the noise is severer.
> 2. Directly adopting class-conditioned augmentations from the diffusion model can slightly rise the prediction accuracy, as also evidenced by the generative model augmented classification literature. However, as it does not contain soft-labeled samples, Var. 2 fails to improve model calibration.
>
> Therefore, we can conclude that models are effectively calibrated only when adopting the proper data and re-annotation scheme.

---

### Official Review · Reviewer_xs6o · 2025-03-12

**Overall Recommendation:** 3

**Summary:**

This paper presents Calibration-aware Semantic Mixing (CSM), a novel approach to improving model calibration by generating high-quality augmented data with soft labels. Unlike traditional augmentation methods that rely on one-hot labels, CSM leverages diffusion models to create semantically mixed images with confidence scores. The authors introduce a reannotation strategy based on CLIP features and explore different loss functions, demonstrating that L2 loss leads to better calibration. Experiments on CIFAR-10, CIFAR-100, and Tiny-ImageNet show that CSM surpasses existing calibration techniques.

**Claims And Evidence:**

CSM enhances calibration by generating realistic semantically mixed samples, as evidenced by Figure 1.
Reannotating confidence scores improves performance, which is validated through ablation studies.
L2 loss provides a better balance in learning, leading to improved calibration, supported by both theoretical analysis and empirical results.

**Essential References Not Discussed:**

The paper should consider discussing and comparing its approach with existing post-hoc calibration methods, particularly [1] Test Time Augmentation Meets Post-hoc Calibration, which is closely related to data augmentation for calibration.

**Experimental Designs Or Analyses:**

The experiments primarily focus on ResNet-based models, and additional evaluations on other architectures (e.g., Transformer-based models) would be beneficial to confirm the generalizability of CSM.
Computational overhead is not explicitly discussed—more details on efficiency and resource consumption would enhance the paper.

**Methods And Evaluation Criteria:**

The paper evaluates model calibration using standard metrics, including Expected Calibration Error (ECE) and Adaptive ECE (AECE), across multiple datasets (CIFAR-10, CIFAR-100, Tiny-ImageNet). Additionally, Reliability diagrams and ablation studies are conducted for further comparison.

**Other Comments Or Suggestions:**

N/A

**Other Strengths And Weaknesses:**

Strengths:
Novel method integrating diffusion models for calibration.
Strong empirical results demonstrating superior performance over existing techniques.
Comprehensive evaluation using standard calibration metrics.
Weaknesses:
Limited hyperparameter analysis—the sensitivity of CSM to different configurations is not well explored.
Unclear computational cost—the efficiency trade-offs of using diffusion models for augmentation should be discussed.

**Questions For Authors:**

Since semantic mix augmentation effectively fills sparse regions in the data space and improves local data proximity, could this approach be integrated with [1] to enhance not only calibration but also model robustness? Exploring this synergy could yield further improvements in generalization and uncertainty estimation.

Reference: [1] Proximity-Informed Calibration for Deep Neural Networks

**Relation To Broader Scientific Literature:**

The work builds on existing research in model calibration and data augmentation, presenting a novel approach by incorporating diffusion models for calibration-aware augmentations. However, the discussion could benefit from additional comparisons to post-hoc calibration methods.

**Theoretical Claims:**

The authors provide theoretical justifications for their reannotation strategy and choice of loss function. While the analysis appears rigorous, additional details and proofs in the supplementary material could further strengthen their claims.

---

> ### Author Rebuttal · Authors · 2025-03-31
>
> ## Response to Reviewer xs6o
>
> Thank you for your positive and insightful feedback! Here are our responses:
>
> **Q2-1**: Additional details and proofs in the supplementary material.
>
> **A2-1**:
> Thank you for the kindly comments on the theoretical soundness. The claims made in our paper (including deducted results in Eq.(6)-(10) and Prop. 3.2-3.4) are sufficiently proved in Appendix A. In detail:
> - Eq.(6)-(7) corresponds to Eq.(14)-(15) with
>   - Assumption 1: Classification optimal classifier $\operatorname{E}(\cdot)$ ensures the likelihood ratio of different classes;
>   - Assumption 2: Regarding the feature as the affine set elements of class prototypes with an orthogonal deviation.
> - Eq.(8)-(9) is proved by Eq.(15)-(19).
> - The result of Eq.(10) is acquired from Eq.(9) with the class factor invariance assumption.
> - Proposition 3.4 is first proved through Eq.(22)-(26) with the assumptions given in Definition 3.1.
> - To prove Proposition 3.2 and 3.3, we first give a general analysis of the problem in Line 712-791 (or Eq.(27)-(37)), then prove them by Lemma A.1 and Lemma A.2, respectively. Note that the proof for CE is unconditional, while for FL it's proved with assumption that $\gamma_{FL}=1.0$ and we empirically find FL more imbalanced with larger $\gamma_{FL}$.
>
> These detailed descriptions illustrate the overall framework of the theoretical analysis. We will include connective details and key deductions in the main paper.
>
> **Q2-2**: Additional evaluations on other architectures.
>
> **A2-2**: Our method performs equally or more effectively to others with the Swin-Transformer architecture. Please refer to **A1-3 Table A**.
>
> **Q2-3**: More details on efficiency and resource consumption.
>
> **A2-3**: Thank your for your suggestion. For efficiency analysis, we compare explicitly in terms of the training time as follows:
>
> **Table B:**
> |Methods|CE|MbLS|Mixup|RegMixup|RankMixup|Ours|Ours(EQ-DATA)|
> |-|-|-|-|-|-|-|-|
> |Training Time|2.63h|2.65h|3.48h|3.50h|4.30h|4.28h|2.64h|
>
> One can see the **number of augmented samples per batch** is the **major factor** for training time. CSM outperforms others in ECE/AECE while maintaining reasonable speed. Even with equalized training samples (EQ-DATA, Table 2), it achieves competitive calibration. CSM runs on a single A4000. Augmented samples need no re-generation across model/loss/annotation changes, enabling efficient modular study.
>
> Key details will be added to the main paper.
>
> **Q2-4**: Comparison with [1], a test-time augmentation (TTA) post-hoc calibration method.
>
> **A2-4**: We compare with [1] by evaluating CSM + [1] as follows:
> |Metrics|ECE|AECE|
> |-|-|-|
> |Ours|**1.29**|1.63|
> |Ours+[1]|1.39|**1.53**|
>
> Our integrated method balances ECE and AECE, achieving an optimized AECE of **1.53** on CIFAR-100. Compared to [1] using test-time sample-wise scaling, CSM employs training-time augmentation with inter-sample augmentations to expand the proximity space, enhancing calibration robustness. We will cite [1] and provide full comparisons in the main paper.
>
> **Q2-5**: Computational overhead of diffusion-based augmentation.
>
> **A2-5**: Our analysis in Appendix C shows that CSM requires **few augmented samples** to launch effectively, ensuring low computational costs. Generating augmented sets takes **~4 hours** (CIFAR-10/100) or **~9 hours** (Tiny-ImageNet) on an RTX4090 GPU, comparable to typical training times. Crucially, CSM **eliminates re-generation** when model architectures/parameters change, further enhancing efficiency through its decoupled design. This validates CSM's computational efficiency.
>
> **Q2-6**: Hyperparameter sensitivity analysis.
>
> **A2-6**: We have analyzed parameter $s$ and No. of augmented samples in Appendix C. We analyze $N_{aug}$ (refer to **A4-2 Table C**) and check sensitivity in **A4-2**. Larger $N_{aug}$ generally enhances our method, while within appropriate ranges of other parameters, it yields stable results.
>
> **Q2-7** Possibility to integrate our CSM with [2] to enhance calibration and robustness.
>
> **A2-7**: Thank you for this insightful question. We conduct post-hoc experiments to integrate our method's outputs with [2], acquiring the following result:
> |Errors$\downarrow$|ECE|MCE|AECE|PIECE|
> |-|-|-|-|-|
> |Ours|**1.29**|**0.21**|**1.62**|3.16|
> |Ours+[2]|1.89|0.73|1.82|**3.11**|
>
> Due to limited time, we simply integrate CSM with [2] without further adjustments. Although a simple combination of them doesn't yield superioir ECE/AECE results, we find that the proximity-informed metric PIECE displays better results, which validates the robustness growth related to proximity from the integration. We will cite [2] for analysis.
>
> **References**
>
> [1] Hekler, A., Brinker, T. J., & Buettner, F. (2023, June). Test time augmentation meets post-hoc calibration: uncertainty quantification under real-world conditions. AAAI.
>
> [2] Xiong, M., Deng, A., Koh, P. W. W., Wu, J., Li, S., Xu, J., & Hooi, B. (2023). Proximity-informed calibration for deep neural networks. NeurIPS.

---

### Official Review · Reviewer_TsGd · 2025-03-14

**Overall Recommendation:** 3

**Summary:**

Model calibration typically assumes full certainty in datasets with one-hot labels, limiting accurate uncertainty estimation. To address this, the paper introduces Calibration-aware Semantic Mixing (CSM), a data augmentation framework that synthetically generates diverse training samples annotated with explicit confidence scores using diffusion models. Additionally, the authors propose a calibrated reannotation method and explore suitable loss functions for this new data paradigm. Experimental results show CSM significantly improves model calibration over existing state-of-the-art methods.

## update after rebuttal
Thank you for the author rebuttal. The major concerns regarding clarifications and additional experiments have been addressed. I will maintain my current rating.

**Claims And Evidence:**

- The motivation and necessity of semantic mixing from the perspective of network calibration are well articulated. Additionally, the drawbacks of existing data-driven methods (mixup-based approaches) are clearly defined.

**Essential References Not Discussed:**

Recent state-of-the-art methods, such as ACLS: Adaptive and Conditional Label Smoothing for Network Calibration (ICCV 2023) and Class Adaptive Network Calibration (CVPR 2023), have not been included in the reference list. These works offer significant contributions to network calibration and should be considered for inclusion to provide a more comprehensive and up-to-date overview of current methodologies.

**Experimental Designs Or Analyses:**

- The experimental results across various networks and datasets demonstrate superior performance compared to existing state-of-the-art methods.
-  Although generalization capability is emphasized, experimental validation on larger datasets such as ImageNet and different network architectures such as Transformers appears insufficient. Additionally, comparisons with recent state-of-the-art methods like CALS (CVPR 23) and ACLS (ICCV 23) are missing,.
- The experiment described in Table 2, which compares training times under identical conditions, is commendable, considering the potential increase in training duration due to the diffusion network. The necessity and effectiveness of reannotation are well-demonstrated in Figure 3.
- In the ablation study (lines 382–384), please confirm if the explanations regarding CE and FL are reversed, particularly concerning temperature.
- It would also be beneficial to provide a comparison illustrating the degree of confidence-balancing achieved by using CE, FL, and L2 losses.

**Methods And Evaluation Criteria:**

- Leveraging conditional diffusion models, specifically via a pre-trained diffusion network, to generate semantically mixed images is technically novel within the context of network calibration.
- Further innovation is demonstrated through the identification and resolution of limitations associated with generated labels by introducing a calibration-oriented reannotation process.

**Other Comments Or Suggestions:**

N/A

**Other Strengths And Weaknesses:**

N/A

**Questions For Authors:**

N/A

**Relation To Broader Scientific Literature:**

This approach effectively improves not only the network’s calibration capability but also enhances its interpretability and accuracy.

**Theoretical Claims:**

Further clarification and verification are needed regarding the balanced loss section. Specifically, more clarification on why the proposed L2 loss functions as a balanced loss would be helpful.

---

> ### Author Rebuttal · Authors · 2025-03-31
>
> ## Response to Reviewer TsGd
>
> Thank you for your kind suggestions on clarity and experimental thoroughness. Below are our responses:
>
> **Q1-1**: Clarification on the reason that the proposed L2 loss is a balanced loss.
>
> **A1-2**:
> Thank you for this nice concern. We need to clarify that there exists a **typo in Proposition 3.4** which makes the conclusion confusing. Proposition 3.4 should have been presented as
>
> - $\forall \delta \ge 0$, we have $\beta(p^{L2}_1,p^{L2}_2)=0$,
>
> which is an equation rather than an inequality for the $\mathcal{L}_2$ loss's balance function, meaning that when two similar samples exceed the model's discriminability, $\mathcal{L}_2$ loss tends to balance the learned labels of the harder and softer instances, instead of tending to fit a specific one of them.
>
> Note that the proof of Proposition 3.4 we provided in the Appendix A.3 does prove that $\beta(p^{L2}_1,p^{L2}_2) = 0$. With such theoretical justification, we also present an empirical evidence in **A1-5** regarding the confidence balance score.
>
> Also refer to **A3-1**. We will correct this typo in the revised main paper.
>
> **Q1-2**: Missing comparisons with ACLS [1] and CALS [2].
>
> **A1-2**: We compare our result with theirs in the following tables. The results demonstrate the competitive or superior performance of our method compared to the state-of-the-arts. We will cite these compared methods and include these results in the final paper.
> |ResNet-50|CIFAR-10|||\||Tiny-ImageNet|||
> |-|-|-|-|-|-|-|-|
> |Metrics|ACC|ECE$\downarrow$|AECE$\downarrow$|\||ACC|ECE$\downarrow$|AECE$\downarrow$|
> |ACLS|95.40|1.12|2.87|\||64.84|**1.05**|**1.03**|
> |Ours|**95.79**|**0.54**|**0.33**|\||**66.99**|1.29|1.19|
>
> |ResNet-50|Tiny-ImageNet|||\||ImageNet|||
> |-|-|-|-|-|-|-|-|
> |Metrics|ACC|ECE$\downarrow$|AECE$\downarrow$|\||ACC|ECE$\downarrow$|AECE$\downarrow$|
> |CALS|65.03|1.54|1.38|\||76.44|1.46|**1.32**|
> |Ours|**66.99**|**1.29**|**1.19**|\||**79.87**|**1.32**|1.35|
>
> **Q1-3**: Insufficient validation on ImageNet and Transformers.
>
> **A1-3**: We compare our result with representative methods on ImageNet with the ResNet-50 and Swin-Transformer architectures. Our method performs equally or more effectively compared to these methods, especially to the mixup-based methods. We will include these results in the final paper.
> |ResNet50|ImageNet|||
> |-|-|-|-|
> |Metrics|ACC|ECE$\downarrow$|AECE$\downarrow$|
> |CE|73.96|9.10|9.24|
> |Mixup|75.84|7.07|7.09|
> |CRL|73.83|8.47|8.47|
> |MbLS|75.39|4.07|4.14|
> |RegMixup|75.64|5.34|5.42|
> |RankMixup|74.86|3.93|3.92|
> |CALS|76.44|1.46|**1.32**|
> |Ours|**79.87**|**1.32**|1.35|
>
> **Table A:**
> |SwinTransformerV2|ImageNet|||
> |-|-|-|-|
> |Metrics|ACC|ECE$\downarrow$|AECE$\downarrow$|
> |CE|75.60|9.95|9.94|
> |LS|75.42|7.32|7.33|
> |FL|75.60|3.19|3.18|
> |FLSD|74.70|2.44|2.37|
> |MbLS|77.18|1.95|1.73|
> |CALS|77.10|1.61|**1.69**|
> |Ours|**81.08**|**1.49**|1.86|
>
> **Q1-4**: Potential reversal of the following CE/FL explanations in ablation (lines 382–384).
> > "In contrast, CE and FL losses often require temperature adjustments, with CE favoring sharper labels and FL for softer ones, aligning with our theoretical expectations from Section 3.3."
>
> **A1-4**: The analysis corresponds to the searched temperature values of Mixup and CSM in Table 4, where CE results sometimes involve a searched temperature larger than 1.0 compared using Mixup (Mixup (**CE**): **T = 1.3**), while FL results produce a searched temperature of **T = 0.9 < 1** compared with our CSM (CSM (**FL**)). These two specific results highlight the nature of CE and FL losses.
>
> As studied by existing works, a higher post-temperature can indicate the model's over-confidence while a lower one suggests under-confidence of the pre-temperature model. Therefore, with soft labels during training, these phenomena indicate a preference/bias of fitting different samples for the adopted losses, *i.e.*, harder labels (*e.g.*, close to one hot) vs. softer labels (*e.g.*, mixup labels with $\lambda=0.5$).
>
> **Q1-5**: Confidence-balancing comparison across CE, FL, and L2.
>
> **A1-5**: We explicitly compute the average balance scores to illustrate the confidence balancing results here:
> |Loss Objectives|CE|Our Loss|FL|
> |-|-|-|-|
> |CIFAR-100|-0.1438|-0.1330|-0.0393|
> |Relative Value|**-0.0108**|**0.0000**|**+0.0937**|
>
> Our loss shows a clear confidence balance between CE and FL, confirming its effectiveness, though empirical values are typically negative due to the rareness of indistinguishable pairs and easier learning of high-confidence samples in practical experiments.
>
> **References**
>
> [1] Park, H., Noh, J., Oh, Y., Baek, D., & Ham, B. (2023). Acls: Adaptive and conditional label smoothing for network calibration. In Proceedings of the IEEE/CVF International Conference on Computer Vision (pp. 3936-3945).
>
> [2] Liu, B., Rony, J., Galdran, A., Dolz, J., & Ben Ayed, I. (2023). Class adaptive network calibration. In Proceedings of the IEEE/CVF conference on computer vision and pattern recognition (pp. 16070-16079).

---

### Decision · Program_Chairs · 2025-05-01

**Decision:**

Accept (poster)

**Comment:**

The paper got three weak accepts and an accept. Among the important concerns raised by reviewers in the pre-rebuttal stage are:

- limited hyperparameter analyses
- comparison with some recent calibration methods
- clarification on a theoretical claim
- limited discussion on computational efficiency
- missing experiments with transformer model

All reviewers acknowledged the rebuttal and decided to keep their original (pre-rebuttal) rating of either weak accept or accept. Given that the rebuttal provides adequate responses to reviewer's comments and no major concern remain, the decision is to recommend acceptance of the paper. Authors are recommended to include important reviewer's comments in the final version.